# A robust method for measuring aminoacylation through tRNA-Seq

Kristian Davidsen[1,2], Lucas B Sullivan[1]*

[1]Human Biology Division, Fred Hutchinson Cancer Research Center, Seattle, United States; [2]Molecular and Cellular Biology Program, University of Washington, Seattle, United States

**Abstract** Current methods to quantify the fraction of aminoacylated tRNAs, also known as the tRNA charge, are limited by issues with either low throughput, precision, and/or accuracy. Here, we present an optimized charge transfer RNA sequencing (tRNA-Seq) method that combines previous developments with newly described approaches to establish a protocol for precise and accurate tRNA charge measurements. We verify that this protocol provides robust quantification of tRNA aminoacylation and we provide an end-to-end method that scales to hundreds of samples including software for data processing. Additionally, we show that this method supports measurements of relative tRNA expression levels and can be used to infer tRNA modifications through reverse transcription misincorporations, thereby supporting multipurpose applications in tRNA biology.

*For correspondence:
lucas@fredhutch.org

Competing interest: The authors declare that no competing interests exist.

## eLife assessment

This **valuable** paper presents a new protocol for quantifying tRNA aminoacylation levels by deep sequencing. The improved methods for discrimination of aminoacyl-tRNAs from non-acylated tRNAs, more efficient splint-assisted ligation to modify the tRNAs' ends for the following RT-PCR reaction, along with the use of an error-tolerating mapping algorithm to map the tRNA sequencing reads provide new tools for anyone interested in tRNA concentrations and functional states in different cells and organisms. The results and conclusions are **solid**, with well-designed tests to optimize the protocol under different conditions.

## Introduction

Quantification of transfer RNA (tRNA) aminoacylation, also referred to as charge, has been performed using radiolabeling (*Wolfson and Uhlenbeck, 2002*), amino acid analysis (*Hill and Struhl, 1986*), northern blotting (*Ho and Kan, 1987*; *Varshney et al., 1991*; *Stenum et al., 2017*), DNA microarrays (*Dittmar et al., 2005*), and high-throughput sequencing (*Evans et al., 2017*). While radiolabeling is highly accurate, it is limited to purified tRNAs undergoing lab manipulation. Northern blotting uses differential migration of acylated tRNA during electrophoresis to measure acylation levels but has many known limitations such as cross-binding probes, low sensitivity, low throughput on multiple tRNAs, insufficient band separation, etc. Chemical differentiation of acylated tRNAs combined with DNA microarrays were introduced to circumvent the problems with northern blotting, but has since been superseded by high-throughput sequencing approaches that enable quantification on all tRNAs in one experiment.

Chemical differentiation of acylated tRNAs is achieved using the Malaprade reaction to attack the 2,3-dihydroxyls on the 3' ribose of deacylated tRNA, causing ring opening and destabilization. The destabilized base is then eliminated using high pH and heat, resulting in a one base truncated 3' sequence of uncharged tRNAs compared to those protected by aminoacylation. This sequence of

reactions was characterized and used extensively in the past in an effort to sequence RNA molecules (*Whitfeld and Markham, 1953*; *Whitfeld, 1954*; *Khym and Cohn, 1961*; *Neu and Heppel, 1964*), and while futile for RNA sequencing, the single-base truncation has proven highly useful to 'tag' deacylated tRNAs. We shall refer to this reaction sequence as the 'Whitfeld reaction' (*Figure 1—figure supplement 1*).

The accuracy and robustness of aminoacylation measurements depend on two parts: the completeness of the Whitfeld reaction and the quality of tRNA sequencing (tRNA-Seq). A major problem in tRNA-Seq is base modifications, known to be numerous on tRNAs. These can lead to stalling, misincorporation, skipping, or falloff during the reverse transcription (RT) step of the sequencing protocol (*Motorin et al., 2007*). The RT polymerase is most severely affected by base modifications disrupting the Watson-Crick base pairing, while other modifications are often less impactful or silent (*Wang et al., 2021*; *Sas-Chen and Schwartz, 2019*). To increase RT readthrough the demethylase AlkB has been used (*Zheng et al., 2015*; *Cozen et al., 2015*), while more recently optimization of incubation conditions, including low salt and extended incubation time, can similarly increase readthrough (*Behrens et al., 2021*). Several other factors can also lead to errors in tRNA-Seq such as low RNA integrity, incomplete deacylation prior to adapter ligation, adapter ligation bias, PCR amplification bias, and errors in read alignment, necessitating further protocol optimization to overcome these issues.

Adapter ligation bias is another well-documented problem in small-RNA sequencing (*Fuchs et al., 2015*; *Zhuang et al., 2012*), but receives little attention in most tRNA-Seq protocols where it is particularly problematic because adapters often incorporate a barcode for sample multiplexing. The problem is further exacerbated when tRNA-Seq is coupled with the Whitfeld reaction, because this creates different sequence contexts for ligation of aminoacylated and deacylated tRNAs. One solution is to optimize conditions such that the ligation goes to completion. To that end, the tRNA secondary structure provides a useful opportunity as it contains four nucleotides on the 3' end that do not participate in the base pairing of the acceptor stem. These are the discriminator bases followed by the invariant CCA-end (*Figure 1*). These free nucleotides can be engaged in base pairing by an oligo splint designed to guide the ligation of the adapter and can improve tRNA specificity and ligation efficiency (*Shigematsu et al., 2017*; *Smith et al., 2015*).

Read mapping is another known problem for tRNA-Seq. It arises due to the high error rate of the RT polymerase when reading through modified bases in addition to frequent falloff. In combination, reads will often not have any continuous stretch of more than 15 nt that perfectly match its reference. This is a problem for almost all alignment algorithms because they rely on some variation of subsequence matching to enable speed-up. The problem has been addressed by clustering of the reference sequences (*Hoffmann et al., 2018*) as well as masking known modified positions in the reference sequences (*Behrens et al., 2021*).

In recent years many variations of the tRNA-Seq method have been published (*Wang et al., 2021*; *Zheng et al., 2015*; *Cozen et al., 2015*; *Shigematsu et al., 2017*; *Erber et al., 2020*; *Thomas et al., 2021*; *Lucas et al., 2024*; *Pinkard et al., 2020*; *Warren et al., 2021*; *Yamagami and Hori, 2023*), but only a few couple it with the Whitfeld reaction to probe aminoacylation levels (*Evans et al., 2017*; *Behrens et al., 2021*; *Watkins et al., 2022*) and little is known about the precision and accuracy of these measurements. Here, we present an up-to-date method for charge tRNA-Seq that integrates new and existing developments, including improved Whitfeld reaction chemistry, splint-assisted ligation, high readthrough RT-PCR, and improved read mapping, enabling us to measure tRNA charge, expression, and modifications (*Figure 1*). We perform tests of the quantitative capabilities of the method and determine its precision and accuracy. Finally, we provide an open-source code repository, enabling others to use our read processing, mapping, and statistical tools on their own data (https://github.com/krdav/tRNA-charge-seq, copy archived at *Davidsen, 2024*).

## Results
### Optimizing the Whitfeld reaction for charge tRNA-Seq

The use of periodate oxidation to discriminate aminoacylated tRNA by sequencing was first used by *Dittmar et al., 2005*, for microarray measurements and then elegantly adapted to high-throughput sequencing by *Evans et al., 2017*. However, we found noticeable differences between the conditions

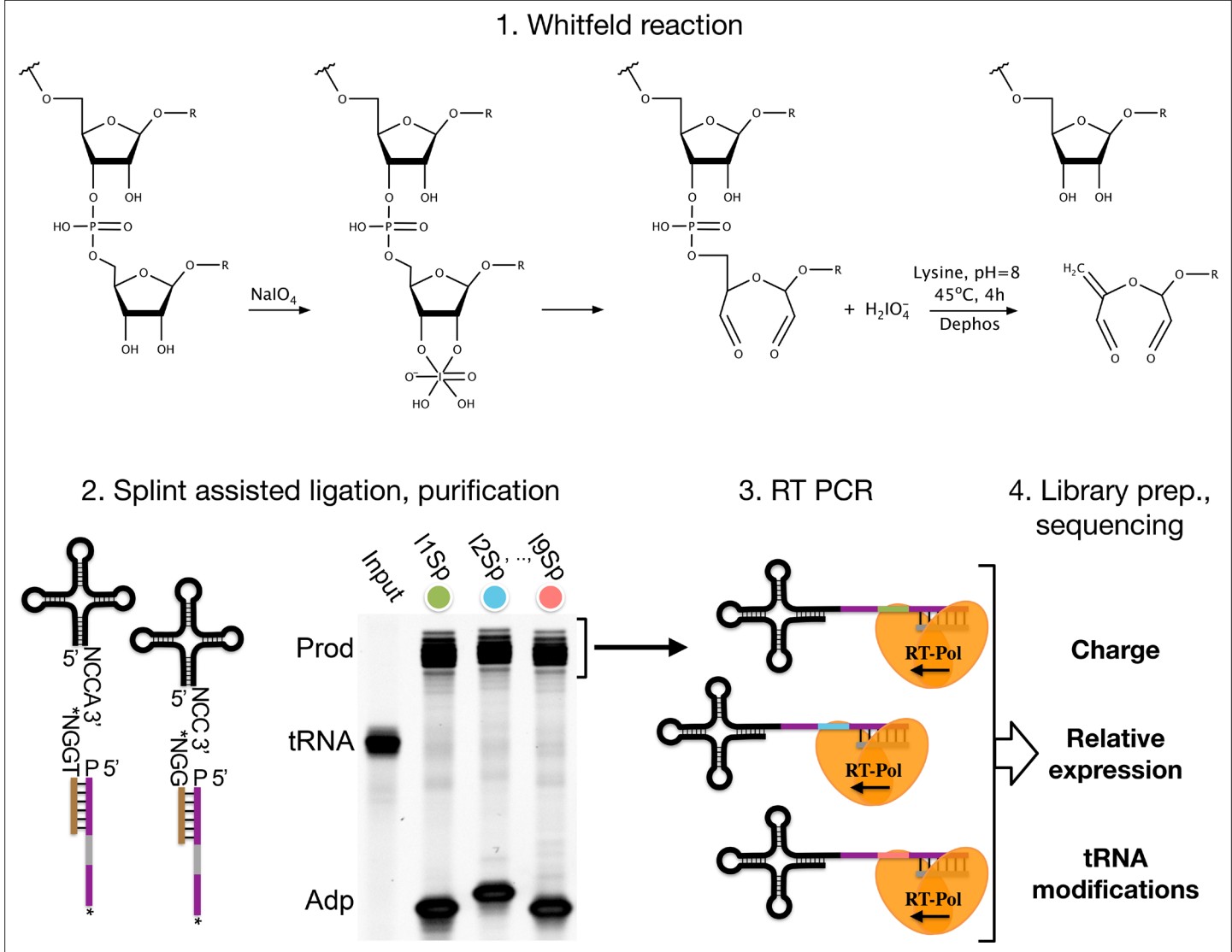

**Figure 1.** Summary illustrating the steps of the charge transfer RNA sequencing (tRNA-Seq) method we used to measure aminoacylation, relative expression, and tRNA modification levels. First, the Whitfeld reaction (detailed in *Figure 1—figure supplement 1*) is used to discriminate between tRNAs with and without an aminoacylation by cleaving off the 3' base of deacylated tRNA. Second, the tRNA secondary structure exposes the discriminator base (N) followed by the CCA/CC-end, creating a sticky end for splint-assisted ligation to a barcoded adapter. P at the 5' end of adapter oligos indicates phosphorylation. Stars (*) on the 3' end of splint and adapter oligos indicate modifications to block self-ligation. Third, using the purified ligation product, RT-PCR is used to generate cDNA. Fourth, the cDNA is converted into a dsDNA library and sequenced to determine tRNA charge, expression, and modifications.

The online version of this article includes the following figure supplement(s) for figure 1:

**Figure supplement 1.** Schematic of the Whitfeld reaction with acylated and deacylated transfer RNA (tRNA) leading to generation of CCA and CC-ending tRNAs.

reported optimal for periodate oxidation in biochemical assays in the past (*Khym and Cohn, 1961*; *Neu and Heppel, 1964*; *Khym and Uziel, 1968*; *Dyer, 1956*) and those used in charge tRNA-Seq today (*Evans et al., 2017*; *Behrens et al., 2021*; *Watkins et al., 2022*; *Pavlova et al., 2020*; *Tsukamoto et al., 2022*). We therefore reasoned that it would be valuable to find a set of optimal conditions for the Whitfeld reaction when applied to charge tRNA-Seq. To do this, we used an *E. coli* tRNA-Lys-CCA oligo and measured conversion to its 1 nt truncated product.

Periodate oxidation of cis-glycols is known to occur rapidly, even at low temperature (*Dyer, 1956*); therefore, we tested if oxidation could be performed on ice to protect tRNA aminoacylations prone to hydrolysis. We found that complete oxidation is achieved after just 5 min (*Figure 2*, panel A) and

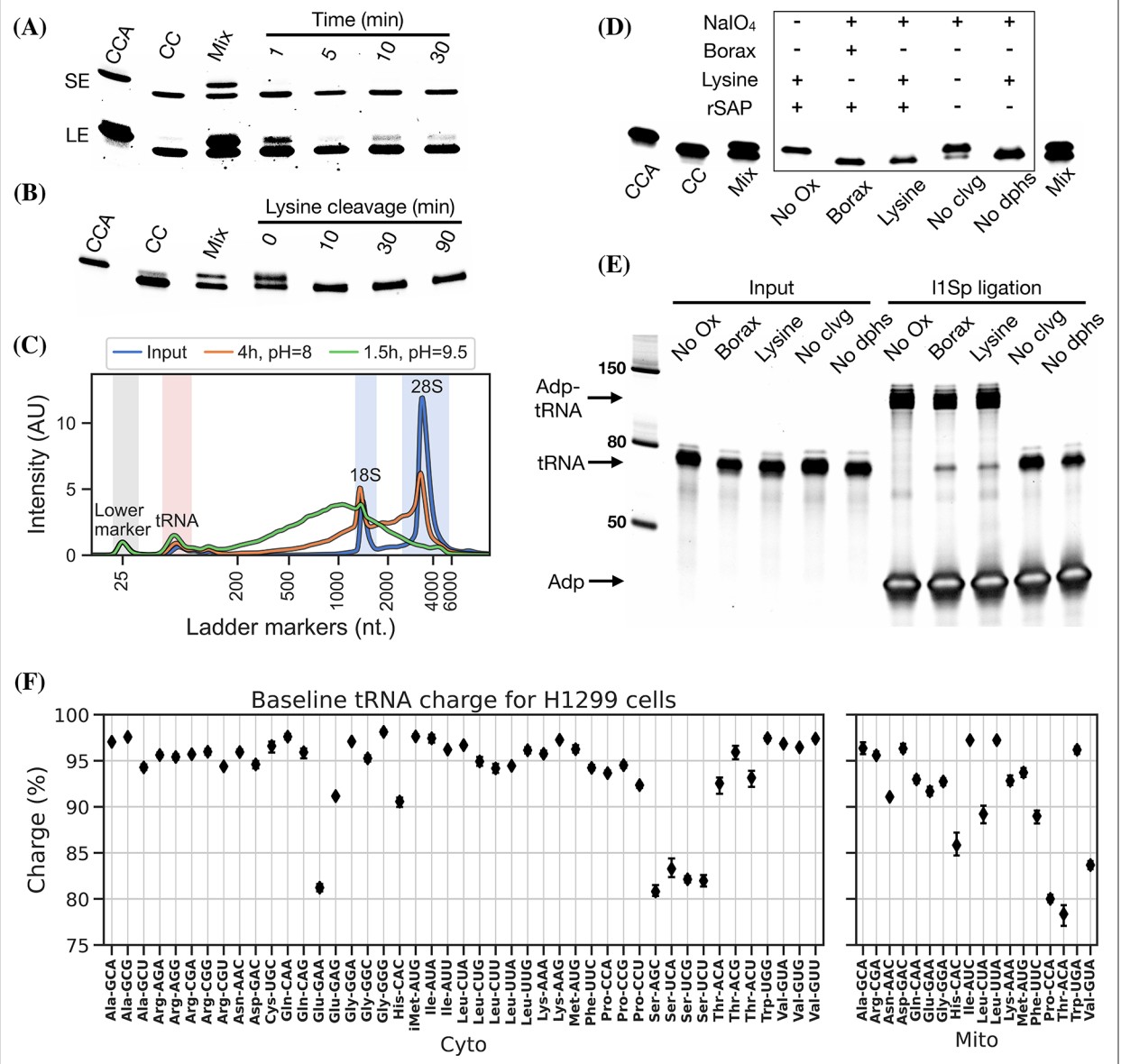

**Figure 2.** Optimizing the chemistry of charge transfer RNA sequencing (tRNA-Seq). (**A**) Time required to complete periodate oxidation of the *E. coli* tRNA-Lys-CCA oligo on ice. Following oxidation, RNA was processed similar to *Evans et al., 2017*, to cleave off the 3′ adenosine. Successful cleavage produce *E. coli* tRNA-Lys-CC. CCA, input oligo. CC, product oligo. Mix, 50/50 mix of CCA and CC. SE, short exposure. LE, long exposure. (**B**) Time required to complete lysine cleavage of the *E. coli* tRNA-Lys-CCA oligo (CCA) at 45°C, pH = 8. Cleavage at time 0 is likely due to the heat denaturation step performed in RNA loading buffer prior to running the gel. (**C**) TapeStation electropherogram comparing stability of whole-cell RNA before and after 4 hr lysine cleavage at pH = 8 or 1.5 hr borax cleavage at pH = 9.5. tRNA range marked by red background, 18/28S by blue. See *Figure 2—figure supplement 1*, panel B for RNA stability timecourse as it occurs on a gel. (**D**) Effect of individual components on cleavage of the *E. coli* tRNA-Lys-CCA oligo (CCA). All samples were processed as a one-pot reaction, except the borax sample which was processed similar to *Evans et al., 2017*. rSAP, shrimp alkaline phosphatase; No Ox, no periodate oxidation; No clvg, no lysine cleavage; No dphs, no dephosphorylation. (**E**) Ligation test comparing the effect of RNA processing. Deacylated and gel-purified human tRNA was processed identically as in panel (**D**), then ligated to adapter l1Sp. Other adapters were tested with similar results (*Figure 2—figure supplement 5*, panel A). (**F**) Baseline tRNA aminoacylation charge in H1299 cells grown in Dulbecco's Modified Eagle's Medium (DMEM) (four replicates, bootstrapped 95% confidence interval of the mean). Only highly expressed codons are shown (more than 1000 reads per million). Charge on tRNA[His] is possibly erroneously low because the discriminator base is shielded by base pairing (*Heinemann et al., 2012*), creating a steric hindrance for the splint-assisted ligation.

The online version of this article includes the following source data and figure supplement(s) for figure 2:

**Source data 1.** Numeric source data for *Figure 2*.

**Source data 2.** Original files for images in *Figure 2*.

*Figure 2 continued on next page*

*Figure 2 continued*

**Source data 3.** Annotated uncropped images used in *Figure 2*.

**Figure supplement 1.** Optimizing lysine-induced cleavage for the charge transfer RNA sequencing (tRNA-Seq) method.

**Figure supplement 1—source data 1.** Numeric source data for *Figure 2—figure supplement 1*.

**Figure supplement 1—source data 2.** Original files for images in *Figure 2—figure supplement 1*.

**Figure supplement 1—source data 3.** Annotated uncropped images used in *Figure 2—figure supplement 1*.

**Figure supplement 2.** Measurement bias in charge transfer RNA sequencing (tRNA-Seq) using blunt-end ligation.

**Figure supplement 2—source data 1.** Numeric source data for *Figure 2—figure supplement 2*.

**Figure supplement 3.** Despite optimization attempts, high ligation efficiency could not be achieved for blunt-end ligation.

**Figure supplement 3—source data 1.** Original files for images in *Figure 2—figure supplement 3*.

**Figure supplement 3—source data 2.** Annotated uncropped images used in *Figure 2—figure supplement 3*.

**Figure supplement 4.** Ligation efficiency of all the barcoded adapters is high and depends on splint complementarity.

**Figure supplement 4—source data 1.** Original files for images in *Figure 2—figure supplement 4*.

**Figure supplement 4—source data 2.** Annotated uncropped images used in *Figure 2—figure supplement 4*.

**Figure supplement 5.** Ligation test comparing the effect of RNA processing.

**Figure supplement 5—source data 1.** Original files for images in *Figure 2—figure supplement 5*.

**Figure supplement 5—source data 2.** Annotated uncropped images used in *Figure 2—figure supplement 5*.

**Figure supplement 6.** RT readthrough comparing TGIRT to Maxima.

**Figure supplement 6—source data 1.** Numeric source data for *Figure 2—figure supplement 6*.

**Figure supplement 6—source data 2.** Original files for images in *Figure 2—figure supplement 6*.

**Figure supplement 6—source data 3.** Annotated uncropped images used in *Figure 2—figure supplement 6*.

**Figure supplement 7.** Transfer RNA (tRNA) expression level measurements are sensitive to the choice of reverse transcription (RT) polymerase while charge measurements are not.

**Figure supplement 7—source data 1.** Numeric source data for *Figure 2—figure supplement 7*.

**Figure supplement 8.** Charge transfer RNA sequencing (tRNA-Seq) control samples and spike-ins validate the method.

**Figure supplement 8—source data 1.** Numeric source data for *Figure 2—figure supplement 8*.

**Figure supplement 9.** Charge measurement comparison between H1299 at baseline (this work) and 293T at baseline (*Evans et al., 2017*).

**Figure supplement 9—source data 1.** Numeric source data for *Figure 2—figure supplement 9*.

**Figure supplement 10.** tRNA homology requires careful PCR conditions.

**Figure supplement 10—source data 1.** Original files for images in *Figure 2—figure supplement 10*.

**Figure supplement 10—source data 2.** Annotated uncropped images used in *Figure 2—figure supplement 10*.

therefore chose 10 min as optimal, with incubation on ice and in the dark because sunlight induces periodate oxidation side reactions (*Erskine et al., 1953*).

Oxidation of deacylated tRNA yields a dialdehyde on the terminal ribose which enables the phosphoric ester linkage to be broken in a β-elimination reaction (*Rammler, 1971*; *Uziel, 1973*), yielding an unsaturated product (*Figure 1—figure supplement 1*). While this cleavage reaction is complex, involving several semi-stable intermediates and different pathways depending on the pH, it appears to be induced by high pH and the presence of a primary amine (*Uziel, 1975*). Lysine has been identified as a good source of primary amine and incubation at 45°C has been found optimal (*Khym and Cohn, 1961*; *Neu and Heppel, 1964*). In previous charge tRNA-Seq methods, a borax buffered solution at pH = 9.5 has been used to induce cleavage, instead we wanted to test using lysine at pH = 8 to improve RNA stability. We found complete cleavage after just 10 min (*Figure 2*, panel B); however, this step also serves as deacylation step and some aminoacylations were still measurable after up to 90 min of lysine cleavage (*Figure 2—figure supplement 1*, panel A). Therefore, we settled on a 4 hr incubation time, but even with this extended incubation, the decrease in pH made a large improvement on RNA integrity (*Figure 2*, panel C).

Finally, we wanted to perform the Whitfeld reaction as a one-pot reaction as shown by *Watkins et al., 2022*. However, we found that the typical quenchers used to remove unreacted periodate (glucose or ribose) are not compatible with lysine-induced cleavage (*Figure 2—figure supplement 1*,

panel C). This is likely due to the generation of dialdehydes that cross-link lysines; therefore, we chose to use ethylene glycol which forms formaldehyde upon periodate quenching. Additionally, ethylene glycol reacts fast and can be added in high molar excess without negatively affecting subsequent steps, thus enabling the whole Whitfeld reaction in one tube (*Figure 2*, panel D).

### Blunt-end adapter ligation can introduce charge measurement bias

Following the Whitfeld reaction tRNAs must be sequenced in order to measure aminoacylation levels. To achieve this with enough throughout, we chose to ligate the samples to barcoded adapters to enable sample pooling before the RT-PCR step (*McGlincy and Ingolia, 2017*). However, we found that the measured charge was highly variable between replicates and that the measurements were biased by the barcode identity to an unacceptable degree (*Figure 2—figure supplement 2*). We hypothesized that this is due to ligation bias commonly encountered in blunt-end ligation (*Fuchs et al., 2015*; *Zhuang et al., 2012*; *Jayaprakash et al., 2011*) and reasoned that increasing ligation efficiency could mitigate the bias. Our attempts to improved ligation efficiency failed as we were never able to reach more than ~50% ligation of the input tRNA (*Figure 2—figure supplement 3*); however, we note that others have reported high blunt-end ligation efficiency using an optimized adapter design (*Behrens et al., 2021*).

### Splint-assisted ligation improves efficiency

Inspired by *Smith et al., 2015*, and *Shigematsu et al., 2017*, we turned to splint-assisted ligation. This approach utilizes that tRNAs have four nucleotides protruding from the 3' end and therefore available for base pairing: the discriminator base, which can be any of the four RNA nucleotides, followed by the invariant CCA-end. The splint oligo is designed to bind both the 3' end of tRNAs and the 5' end of an adapter (*Figure 1*), thus bringing the two into proximity and increasing ligation efficiency. However, whereas earlier uses of splint-assisted ligation could assume that all tRNAs end on CCA, we have a mix of CCA and CC-ending tRNAs and therefore needed to use two splints. As tRNAs compete for ligation it is imperative that CCA-ending tRNAs, with stronger interaction with the splint, is not favored over CC-ending tRNAs. Fortunately, we observed a near-complete ligation between all of our nine barcoded adapters and both CCA-ending human tRNA and a CC-ending *E. coli* tRNA-Lys oligo (*Figure 2—figure supplement 4*, panels A and B). The ligation was specific as it was fully dependent on complementarity between the tRNA and the splint (*Figure 2—figure supplement 4*, panel C). As we are only interested in ligation between tRNA and adapter, we block all other possible ligations through dephosphorylation of the 5' tRNA nucleotide and oligo modifications blocking the 3' end of adapter and splint oligos. This affords us the advantage of using a pure DNA splint without any RNA nucleotides as those used in previous publications (*Smith et al., 2015*; *Shigematsu et al., 2017*; *Pinkard et al., 2020*; *Warren et al., 2021*; *Thomas et al., 2021*; *Lucas et al., 2024*).

Importantly, we validated that tRNA processed using the one-pot Whitfeld reaction could be effectively used as substrate in the ligation reaction (*Figure 2*, panel E and *Figure 2—figure supplement 5*, panel A). We noted that a small amount of unligated tRNA appeared in reactions with tRNA oxidized with periodate. This unligated tRNA is of unknown origin and largely refractory to further ligation (*Figure 2—figure supplement 5*, panel B); however, as shown later using charge titration, this did not have a measurable impact on the accuracy of the aminoacylation measurement.

### Combining optimizations results in a robust method for measuring tRNA charge

After combining the optimized Whitfeld reaction with subsequent splint-assisted ligation, we used the RT-PCR method proposed by *Behrens et al., 2021*, using the TGIRT polymerase (*Mohr et al., 2013*) to maximize the readthrough of modified nucleotides. We later found that similar readthrough could be achieved using Maxima RT polymerase (*Figure 2—figure supplement 6*), albeit with worse mapping statistics (*Supplementary file 4*). Furthermore, while expression level measurements are highly sensitive to the choice of polymerase, charge measurements are not (*Figure 2—figure supplement 7*). The RT-PCR was primed by an oligo containing a 10 nt unique molecular identifier (UMI) to diversify the sequence context for the subsequent circular ligation and allow collapsing of reads derived from the same tRNA molecule during data analysis. A final PCR was performed to attach Illumina barcodes to pool samples for multiplex sequencing.

Using this as our final charge tRNA-Seq method, we use the *E. coli* tRNA-Lys-CCA oligo as a spike-in control before the Whitfeld reaction to validate near-complete conversion to its CC-end product, suggesting efficient periodate oxidation (*Figure 2—figure supplement 8*, panel A). Similarly, we validated the completeness of deacylation using deacylated controls and the integrity of the tRNA CCA-end using non-oxidized controls (*Figure 2—figure supplement 8*, panels B and C). We then measured the baseline charge of H1299 cells grown in Dulbecco's Modified Eagle's Medium (DMEM) using four replicates, observing excellent repeatability and high charge for most codons (*Figure 2*, panel F). Direct comparison to other methods are complicated by the use of different cell lines, RNA extraction, and processing; nevertheless, our charge measurements for H1299 cells at baseline are generally a few percentage points higher than those by *Evans et al., 2017*, for 293T cells (*Figure 2—figure supplement 9*). Interestingly, we also observe low baseline charge for tRNA^Ser codons and a tRNA^Glu codon, validating a similar observation by *Evans et al., 2017*.

## Reference masking improves read mapping

It has previously been noted that alignment of tRNA reads is challenging due to RT misincorporations and falloff (*Hoffmann et al., 2018*; *Behrens et al., 2021*). Most commonly, the Bowtie1 or Bowtie2 aligners have been applied using various settings to accommodate short reads and the many mismatches (*Cozen et al., 2015*; *Zheng et al., 2015*; *Clark et al., 2016*; *Evans et al., 2017*; *Pinkard et al., 2020*). However, while these are ultra-fast and widely used for RNA-Seq, Bowtie1 does not support alignments with insertions or deletions, and although Bowtie2 does, it does not guarantee that the best alignment is returned (*Langmead et al., 2009*; *Langmead and Salzberg, 2012*). We reasoned that many users of tRNA-Seq would rather sacrifice computational speed than mapping accuracy and therefore we apply a full all-against-all local alignment using the Smith-Waterman algorithm to provide the guaranteed best alignment(s). This is possible because the set of tRNA transcripts in a typical species is only a few hundred sequences and thus we are able to align 1e8 tRNA-Seq reads to a human tRNA reference with 457 sequences in less than 8 hr on an Intel Core i7-8700K Processor (12 threads, 4.70 GHz).

In addition to the choice of read alignment method, *Behrens et al., 2021*, found that using an SNP-tolerant alignment substantially improved mapping when modified positions causing RT misincorporations were defined as SNPs and that new modifications could be inferred from the data and used to further improve the alignment. We adapted this approach by masking modified positions in the reference to 'N' while relying solely on the misincorporation information embedded in the sequencing data. We extracted this information using a first-pass alignment and then used mismatch frequencies to pick positions for reference masking. As such, this is an iterative process because the alignment will change slightly with a new masked reference. In addition to the number of iterations, masking is only applied on positions with a minimum mismatch frequency (min_mut_freq) and frequency is calculated either including or excluding reads with multiple transcript alignments (unique_anno_float). Furthermore, a parameter (frac_max_score) controls the sharing of a mask to highly similar transcripts. To find the optimal combination of parameters for reference masking we performed a grid search with the objective of finding the masking that resulted in the least number of reads assigned to transcripts with multiple codons (*Figure 3*, panel A). This lead to 533 positions in the 455 sequence reference getting masked and resulting in an alignment improvement, reducing the reads with multiple codon alignments from 11.71% to 5.09%.

Masked positions do not contribute to the alignment score and thus possibly lowering it below the minimum threshold; however, we observed no trade-off between optimized reference masking and read mapping percentage (*Figure 3*, panel B). Like *Behrens et al., 2021*, we observe a striking difference in the mapping of certain tRNA transcripts with inosine at the first position of the anticodon (position 34; I34), e.g., transcripts decoding the Ser-UCU codon (IGA anticodon) (*Figure 3*, panel C). Generally, reference masking appears to increase annotations for around a dozen transcripts but a substantial mapping change only occurs for six codons (*Figure 3—figure supplement 1*, panel A). The effect of reference masking on the charge measurements was low as expected because this is a relative number (*Figure 3—figure supplement 1*, panel B).

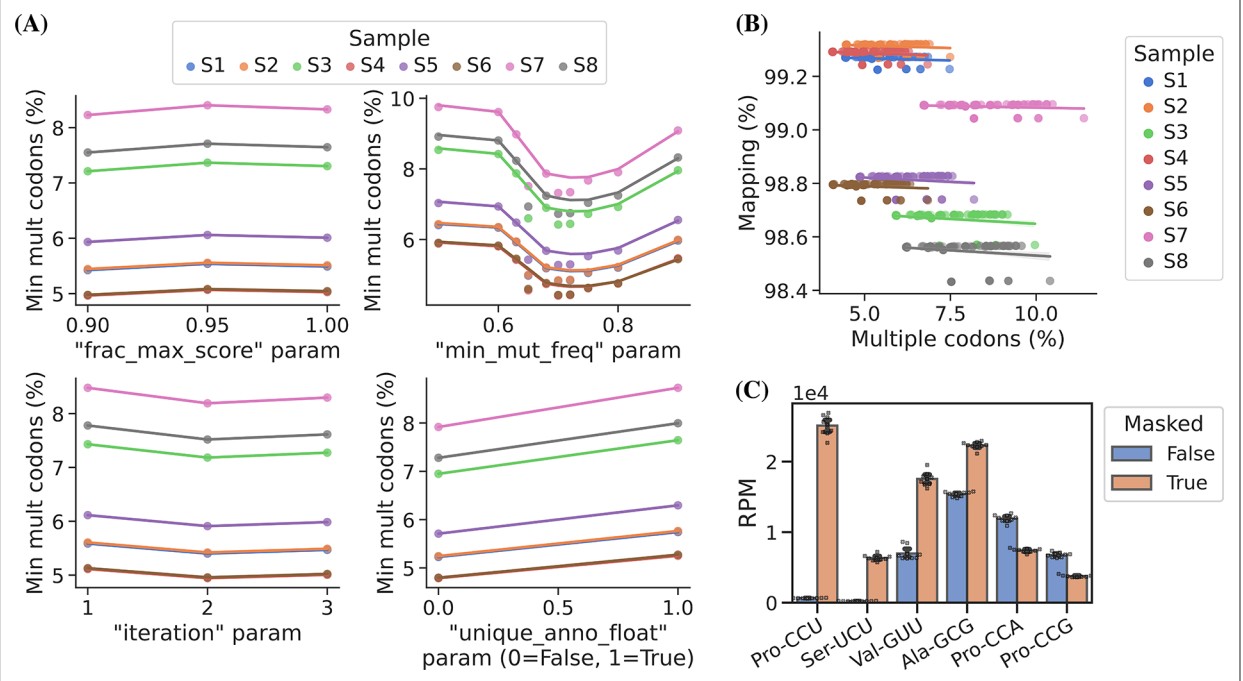

**Figure 3.** Masking of the reference sequences improves alignment performance. (**A**) Grid search optimization of parameters determining the extent of reference masking (see Materials and methods section for details). Each subplot shows the mean effect of one tuning parameter when combined with the combination of all the other three. Parameters used for reference masking are chosen to minimize the percentage of reads assigned to transfer RNAs (tRNAs) with multiple codons. (**B**) There is no trade-off between sequence mapping success and minimizing multiple codon mapping. (**C**) Reference masking increases relative expression levels of select codons. Reads per million (RPM) levels of the codons shown were found before and after optimized reference masking. Error bars are bootstrapped 95% confidence interval of the mean over the nine barcode replicate samples.

The online version of this article includes the following source data and figure supplement(s) for figure 3:

**Source data 1.** Numeric source data for *Figure 3*.

**Figure supplement 1.** Reference masking effect on RPM and charge levels.

**Figure supplement 1—source data 1.** Numeric source data for *Figure 3—figure supplement 1*.

**Figure supplement 2.** Mismatch frequency, gap frequency, and reverse transcription (RT) stop percentage is increased upon periodate oxidation for transcripts known to be 5-methoxycarbonylmethyl-2-thiouridine (mcm$^5$s$^2$U) modified.

**Figure supplement 2—source data 1.** Numeric source data for *Figure 3—figure supplement 2*.

**Figure supplement 3.** Polymerase-dependent mismatch frequency, gap frequency, and reverse transcription (RT) stop percentage for the 5-methoxycarbonylmethyl-2-thiouridine (mcm$^5$s$^2$U) modified position.

**Figure supplement 3—source data 1.** Numeric source data for *Figure 3—figure supplement 3*.

## tRNA modifications are reflected in mismatches, gaps, and RT stops

Our computational method also supports using misincorporation data for inference of nucleotide modifications, which is typically only valid for modifications that disrupt Watson-Crick base pairing such as methylations (*Clark et al., 2016*; *Behrens et al., 2021*). As such the 5-methoxycarbonylmethyl-2-thiouridine (mcm$^5$s$^2$U) modification should be silent; however, thionucleosides are sensitive to periodate treatment, which oxidizes them to sulfonates and makes them sensitive to nucleophilic attack (*Ziff and Fresco, 1968*; *Rao and Cherayil, 1974*). When periodate oxidation of mcm$^5$s$^2$U is followed by lysine cleavage it would presumably result in a lysine adduct (*Ziff and Fresco, 1968*), thus disrupting Watson-Crick base pairing. We verified this by comparing the misincorporation signature in samples processed with/without periodate oxidation, focusing on the human tRNAs Lys-UUU, Gln-UUG, Glu-UUC, and Arg-UCU shown by *Lentini et al., 2018*, to carry the mcm$^5$s$^2$U modification (*Figure 3—figure supplement 2*). Large changes in the misincorporation signature is observed upon periodate oxidation, but curiously some tRNAs respond with a large decrease in RT readthrough while others have an increased mutation and/or gap frequency. Similar observations were recently shown by *Katanski et al., 2022*. Interestingly, the misincorporation signature also depends on the polymerase

used, and thus the Maxima RT polymerase generally shows better readthrough and increased mutation and gap frequency on the mcm$^5$s$^2$U modification compared to TGIRT (*Figure 3—figure supplement 3*).

## Barcode replicates show high precision

To assess measurement precision, we performed our charge tRNA-Seq protocol on the same tRNA sample using all nine barcoded adapters. We used partially deacylated RNA to achieve a representative spread of aminoacylation levels within a single sample (*Figure 4—figure supplement 1*, panel A) and then extracted differences compared to the median barcode replicate measurement. When comparing charge measurements binned by barcode, we observed that most were narrowly distributed with the median close to zero indicating little or no barcode bias (*Figure 4*, panel A). Adapter l4Sp is the exception that proves why barcode bias needs to be investigated, because it is consistently overestimating charge levels, with a median overestimate of ~3 percentage points. Overall however, charge measurements show high precision with a standard deviation from the median of just 1.7 percentage points, with similar results at the transcript level (*Figure 4—figure supplement 2*, panel A).

For reads per million (RPM) values, some barcode replicates were more narrowly distributed than others. However, these differences are small and with a standard deviation from the median of 5.1 percentage points we consider the RPM measurements to be precise (*Figure 4*, panel B and *Figure 4—figure supplement 2*, panel B).

## Charge titration shows high accuracy

Testing the accuracy of charge measurements is a much harder problem. Spiking in a defined ratio of CC and CCA-ending oligo to the ligation reaction is a common approach, but this ignores the possible incompleteness of the Whitfeld reaction. It is also possible to compare to charge measured by northern blotting, but this presents a different set of issues with probe annealing, band resolution, etc. As an alternative, we made a charge titration by mixing different proportions of intact and deacylated RNA allowing us to predict and measure charge levels of over 150 transcripts (*Figure 5*, panel A). The results showed excellent proportionality between predicted and measured charge across the full range of values (*Figure 5*, panel B), thus indicating that the charge measurements are highly accurate. This experiment also confirmed our previous observations that barcode bias is limited to the l4Sp adapter, which is consistently overestimating charge (*Figure 5*, panel C). Additionally, no bias was found in independently prepared sequencing libraries or any of the different mixing proportions of intact and deacylated RNA (*Figure 5—figure supplement 2*).

Inspired by *Evans et al., 2017*, which used radiolabeling techniques to generate a single accurate tRNA charge reference point, we developed a 50% charge control using 3' phosphorylation as protection from periodate oxidation. This control was spiked into samples before the Whitfeld reaction and showed a mean charge of 50.36% and a standard deviation of 1.11 percentage points (*Figure 5—figure supplement 3*, panel B), thus further validating the measurement accuracy of our method.

## Charge tRNA-Seq enables measurement of aminoacylation half-lives of native tRNAs

tRNA aminoacylations are prone to hydrolysis and the effect of pH and temperature on their decay rates has previously been studied (*Hentzen et al., 1972*). Interestingly, *Peacock et al., 2014*, found that the aminoacylation half-life appeared to be determined solely by the identity of the amino acid attachment and not affected significantly by the tRNA sequence or RNA modifications. However, most of the tRNAs used in this study were derived from in vitro transcription and only a limited set of RNA modifications were tested; additionally, the study did not cover all 20 native amino acids. Having developed an accurate method for measuring tRNA charge on over a hundred samples in a single sequencing run, we wanted to use this to determine the aminoacylation half-lives of tRNA transcripts with their native RNA modifications.

We used RNA purified from the H1299 cell line, starting at high tRNA charge (*Figure 2*, panel F), and tracked the aminoacylation decay over time after switching to physiological buffer (pH = 7.2) and incubating at 20°C, similar to *Peacock et al., 2014*. After sampling 11 timepoints with 4 replicates, charge measurements for each transcript were fitted to a first-order decay function to estimate the

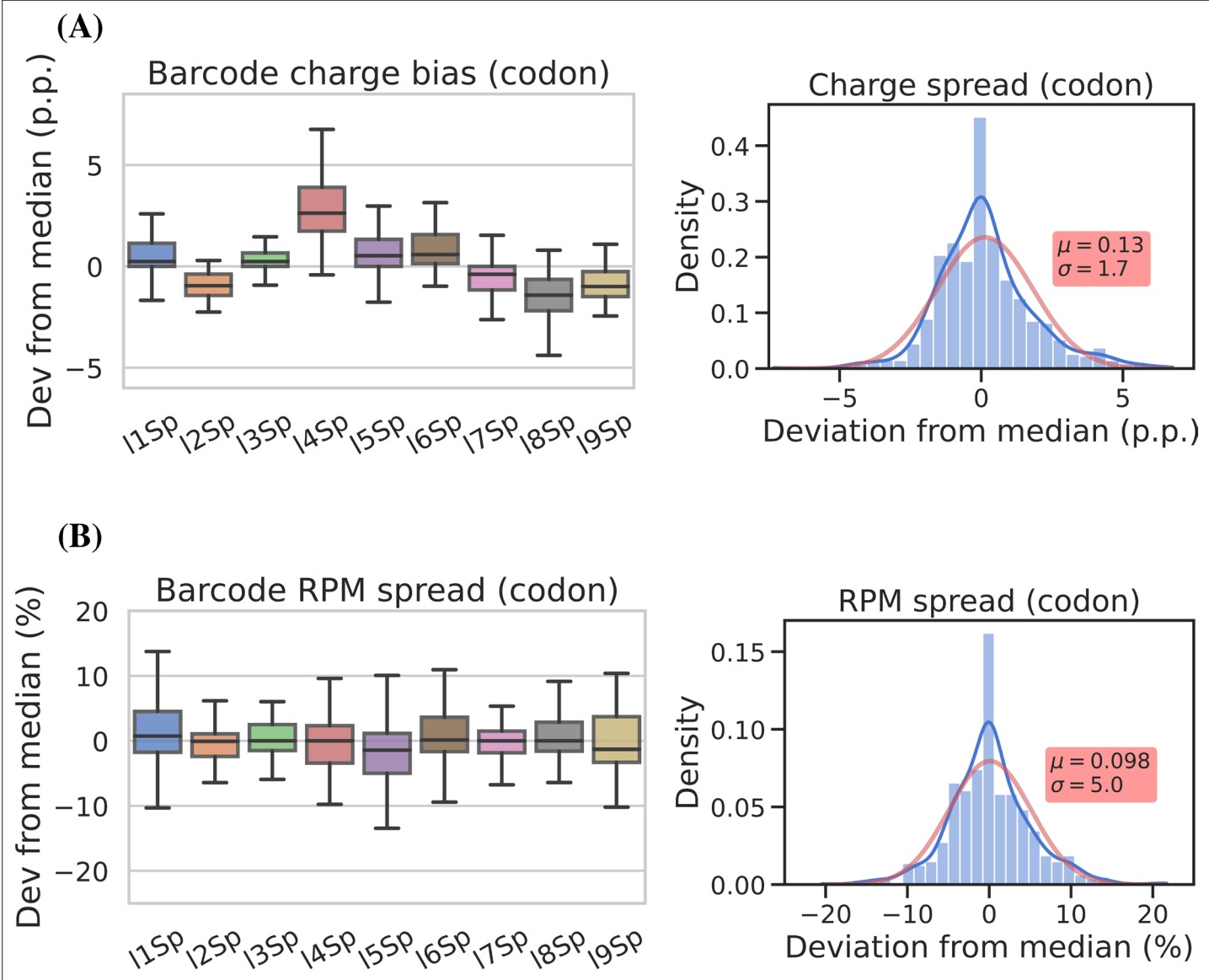

**Figure 4.** Barcode replicates show high precision and limited barcode bias. Each of the nine adapters was ligated to the same sample containing a heterogeneous mix of CC and CCA-ending transfer RNAs (tRNAs). Ligations were then pooled and submitted to the remainder of the charge tRNA sequencing (tRNA-Seq) protocol. (**A**) The percentage point deviation from the median charge at the codon level, grouped by barcode identity (left) or shown summarized as a density plot (right). (**B**) The percentage deviation from the median reads per million (RPM) at the codon level, grouped by barcode identity (left) or shown summarized as a density plot (right). Density plots are provided with kernel density estimate (KDE) in blue, normal distribution estimate in red and inserts with mean (μ) and standard deviation (σ). For plots of transcript level data, see *Figure 4—figure supplement 2*.

The online version of this article includes the following source data and figure supplement(s) for figure 4:

**Source data 1.** Numeric source data for *Figure 4*.

**Figure supplement 1.** Best and worst pairwise comparisons between barcode replicates.

**Figure supplement 1—source data 1.** Numeric source data for *Figure 4—figure supplement 1*.

**Figure supplement 2.** Similar to *Figure 4*, but at the transcript level.

**Figure supplement 2—source data 1.** Numeric source data for *Figure 4—figure supplement 2*.

half-life of each transcripts (*Supplementary file 5*), as exemplified by the representative transcript Lys-TTT-3-1 (*Figure 6*, panel A). When transcripts were grouped by their cognate amino acid, we could confirm that the half-lives are indeed determined mostly by aminoacylation identity and that they span a 37-fold range (*Figure 6*, panel B). Our half-life estimates are highly correlated with those

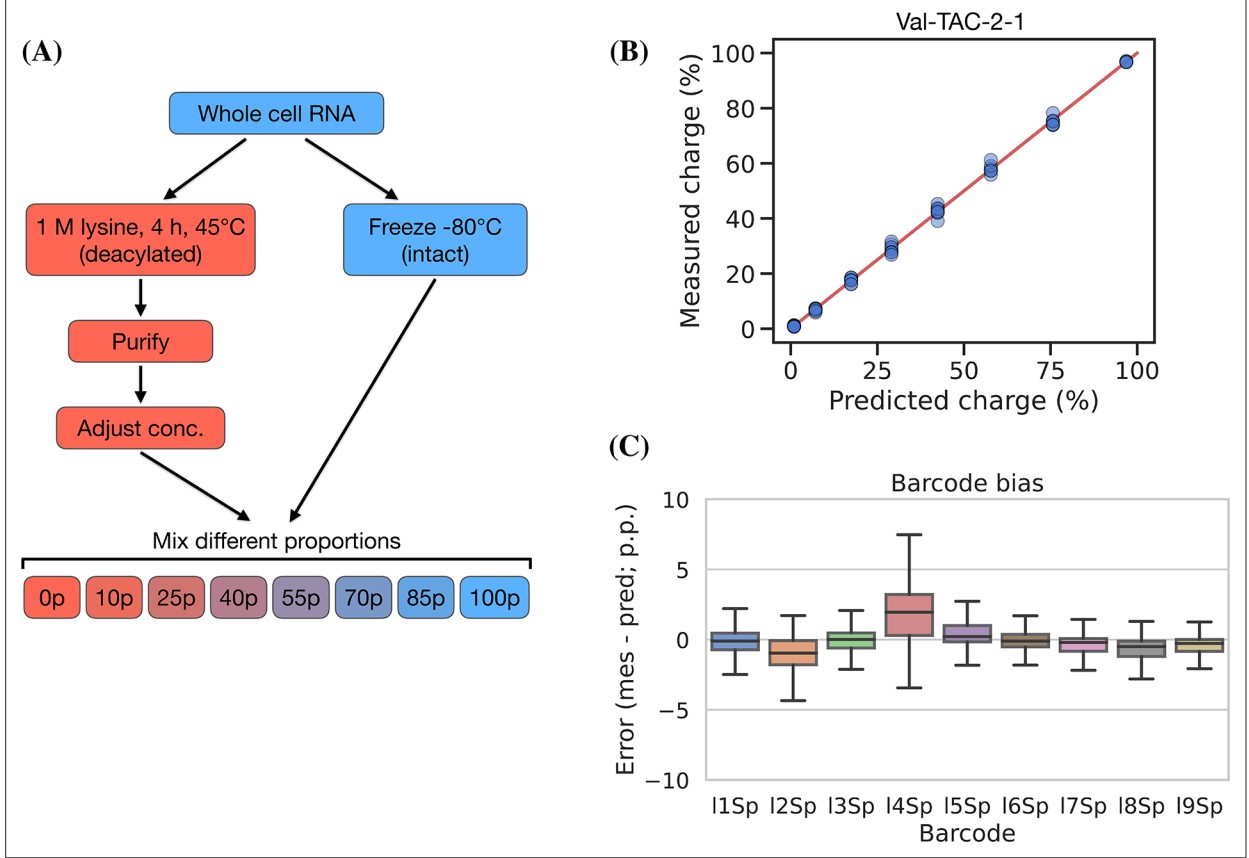

**Figure 5.** Charge titration shows linearity over the full range of charge measurements. (**A**) Schematic illustration of the method to generate samples with predictable charge percentages. (**B**) Titration data for a representative transfer RNA (tRNA) transcript, Val-TAC-2-1, with the red line indicating equality between predicted and measured charge. For reference, the best and worst fitting tRNA transcripts are shown in *Figure 5—figure supplement 1*. (**C**) Error binned by adapter barcode. Error is the percentage point difference between the measured vs. predicted charge for all transcripts in the bin.

The online version of this article includes the following source data and figure supplement(s) for figure 5:

**Source data 1.** Numeric source data for *Figure 5*.

**Figure supplement 1.** The best and worst transcript when ranked based on the sum of squared differences between the measured and predicted charge.

**Figure supplement 1—source data 1.** Numeric source data for *Figure 5—figure supplement 1*.

**Figure supplement 2.** Charge titration prediction error binned by sequencing run and titration sample.

**Figure supplement 2—source data 1.** Numeric source data for *Figure 5—figure supplement 2*.

**Figure supplement 3.** Spike-in control for 50% charge using the *Escherichia coli* tRNA-Thr-CGT oligo.

**Figure supplement 3—source data 1.** Numeric source data for *Figure 5—figure supplement 3*.

**Figure supplement 3—source data 2.** Original files for images in *Figure 5—figure supplement 3*.

**Figure supplement 3—source data 3.** Annotated uncropped images used in *Figure 5—figure supplement 3*.

reported by *Peacock et al., 2014*, but surprisingly ours appear to be approximately fourfold higher despite using the same incubation temperature and a similar buffer, with only slightly lower pH (7.2 vs 7.5; *Figure 6—figure supplement 1*, panel B).

It seems counterintuitive that the aminoacylation half-life should be completely unaffected by the tRNA sequence; however, as the amino acid is attached to the invariant CCA-end, the nucleotides most proximal to the ester bond are the same for all tRNAs. The most proximal non-invariant nucleotide is the discriminator base. Because we sample all transcripts, we are able to observe that the discriminator base is indeed likely to influence the half-life and that a purine base appears to promote a longer aminoacylation half-life than a uracil (*Figure 6*, panel C).

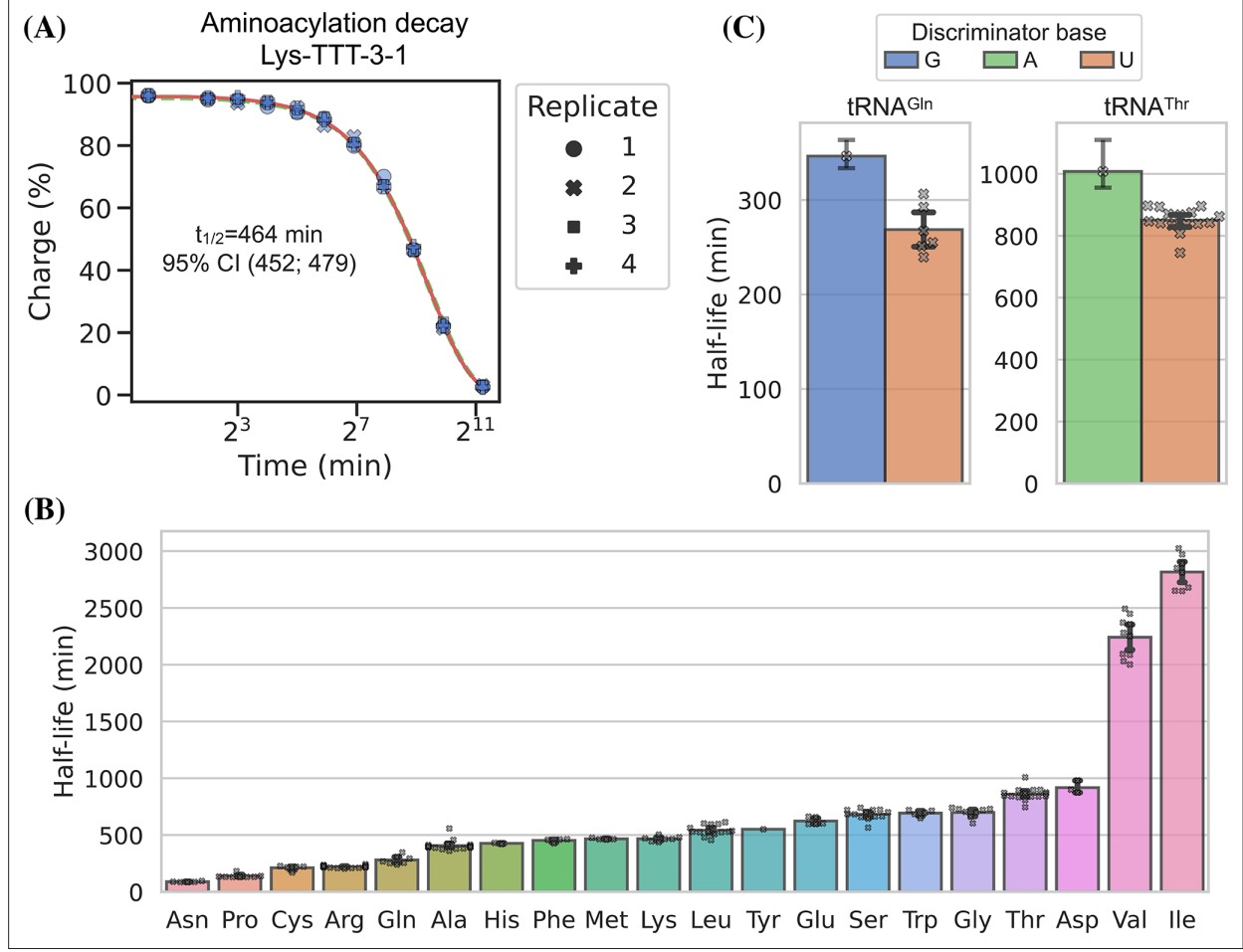

**Figure 6.** Measuring aminoacylation half-life using charge transfer RNA sequencing (tRNA-Seq). (**A**) Aminoacylation decay for a representative tRNA transcript Lys-TTT-3-1 over the 11 timepoints sampled. For reference, the best and worst fitting tRNA transcripts are shown in *Figure 6—figure supplement 2*. The fitted first-order decay to estimate the aminoacylation half-life is shown as a red line. Similar dashed lines are plotted in green for the bootstrapped 95% confidence interval (these are hard to see). (**B**) Aminoacylation half-life estimates grouped by amino acid. Each marker represents one transcript, error bars are bootstrapped 95% confidence intervals of the mean. (**C**) Distribution of aminoacylation half-life estimates for tRNA$^{Gln}$ and tRNA$^{Thr}$ transcripts grouped by discriminator base identity. Error bars are bootstrapped 95% confidence intervals. For the single transcripts with G or A discriminator base the bootstrap is performed on measurement replicates while for the U discriminator base it is performed on the transcript observations.

The online version of this article includes the following source data and figure supplement(s) for figure 6:

**Source data 1.** Numeric source data for *Figure 6*.

**Figure supplement 1.** RNA integrity and comparison to previous half-life values.

**Figure supplement 1—source data 1.** Numeric source data for *Figure 6—figure supplement 1*.

**Figure supplement 1—source data 2.** Original files for images in *Figure 6—figure supplement 1*.

**Figure supplement 1—source data 3.** Annotated uncropped images used in *Figure 6—figure supplement 1*.

**Figure supplement 2.** The best and worst transcript half-life estimates, ranked based on the sum of squared differences between the fitted decay function and the mean charge of the replicates.

**Figure supplement 2—source data 1.** Numeric source data for *Figure 6—figure supplement 2*.

## Discussion

Accurate quantification is a prerequisite for making reliable observations standing the test of time and replication. We have presented a robust method for measuring tRNA charge and extensively validated it in the relevant context of human tRNA. Furthermore, we have quantified the measurement precision of charge and relative expression. Accuracy was only quantified for charge measurement

whereas this is more challenging for expression levels (*Fuchs et al., 2015*). One step toward accurate expression level measurements is efficient adapter ligation, such as the splint-assisted ligation method used herein; however, future versions of tRNA-Seq should strive toward providing better validation and controls for relative expression measurements. In our version of the Whitfeld reaction we use lysine to induce base cleavage at low pH. We have later been made aware that ornithine is an even better inducer of cleavage (*Uziel, 1975*) and thus, the pH of the cleavage reaction could be lowered even further and possibly combined with $Cu^{+2}$ as a deacylation catalyst (*Kroll, 1952*; *Schofield and Zamecnik, 1968*) to shorten incubation times.

In our experience, as well as others (*Shigematsu et al., 2017*), splint-assisted ligation is highly efficient compared to blunt-end ligation. In contrast, *Behrens et al., 2021*, achieved high-efficiency blunt-end ligation, allowing inclusion of non-mature tRNAs without the normal CCA-end. Nevertheless, our results highlight the difficulty of using blunt-end ligations for tRNA-Seq and provide an alternative approach of splint-assisted ligation to help mitigate those issues. One potential issue with our approach is that the tRNA[His] sequence is not ideal for splint-assisted ligation due to the additional G added to the 5' end (*Heinemann et al., 2012*) and thus shielding the discriminator base from base pairing with the splint. Despite this, reads mapping to tRNA[His] are surprisingly abundant and contain both CC and CCA-ends. In future versions of this method, we see the possibility of combining our optimizations with the on-bead sample processing developed by *Watkins et al., 2022*, to eliminate gel purification steps and achieve faster and cleaner processing.

We solve the tRNA alignment problem by non-heuristic alignment which is guaranteed to return the best alignment. This is computationally demanding but nevertheless quite possible on the small number of tRNA transcript references. A more challenging problem is the application of reference masking to improve the annotation accuracy. We used unique codon annotation as the objective in our optimization, but this is a surrogate as the ground truth is unknown. Further improvements could be achieved by simulation of tRNA reads including realistic RT misincorporations, indels, and falloff and optimizing alignment to this simulated ground truth. Additionally, annotation performance could be increased further using tools, such as hidden Markov models, to model complex phenomena such as interaction between modifications (*Wang et al., 2021*; *Hernandez-Alias et al., 2023*).

In summary, we report a robust charge tRNA-Seq method that has been thoroughly tested and validated as precise and accurate for charge measurements.

## Materials and methods
### Cell culture and RNA extraction
The human cell line H1299 was acquired from ATCC and tested to be free from mycoplasma (Myco-Probe, R&D Systems). Cells were maintained in DMEM supplemented with 3.7 g/L sodium bicarbonate, 10% fetal bovine serum, and 1% penicillin-streptomycin solution. Cells were incubated in a humidified incubator at 37°C with 5% $CO_2$.

For RNA extraction, cells were seeded onto a 15 cm dish and grown in DMEM until confluency. The cells were then removed from the incubator, placed on a slope on ice, and media was quickly and thoroughly aspirated before adding 3 mL TRIzol to cover all the cells. From this point onward, everything was kept ice cold to prevent hydrolysis of the aminoacylation. After a 2 min incubation, the cell material was scraped down the slope mixing it with the TRIzol, then 2×1.5 mL was transferred to 2 mL Eppendorf tubes and 0.3 mL chloroform was added. The tubes were vortexed 2 min and then centrifuged (17,000×*g*, 5 min). From each tube, 0.75 mL of the upper layer was transferred to a tube with 0.8 mL isopropanol (IPA), then mixed and incubated 60 min at –20°C. Tubes were then centrifuged (17,000×*g*, 15 min) and RNA pellets were washed twice with 1 mL 80% IPA containing 100 mM sodium acetate (pH = 4.5). These washing steps are critical because TRIzol contains glycerol which will react with and inhibit the subsequent periodate oxidation step. A last wash was performed using 1 mL 100% IPA and after removing the supernatant the RNA pellets were air-dried at room temperature, then stored dry at –80°C.

### Charge tRNA-Seq using blunt-end ligation
For charge tRNA-Seq using blunt-end ligation shown in *Figure 2—figure supplement 2* the protocol described by *Behrens et al., 2021*, was followed with the exception of using different adapter

sequences, a UMI containing RT oligo (*Supplementary file 1*), more rounds of amplification and gel-based size selection for the final sequence library, and using paired-end sequencing. Briefly, whole-cell RNA was extracted, reconstituted in 100 mM sodium acetate (pH = 4.5) and concentration adjusted to 1 µg/µL. A 20 µL sample was moved to a new tube and submitted to periodate oxidation and 3′ base elimination using sodium borate as described by *Evans et al., 2017*. After purification and reconstitution in water, 8 ng of a 50/50 mix of *E. coli* tRNA-Lys-CCA and *E. coli* tRNA-Lys-CC oligo was added as a CCA/CC ratio control. The true ratio of these oligos is hard to control because each contains a different fraction of truncated oligos that will not contribute to the number of mapped reads; however, the sequenced CCA/CC ratio is an important measure of the sample-to-sample variance. Then the RNA was 3′ dephosphorylated using T4 PNK and after another round of RNA purification the tRNA fraction was isolated on a 10% Urea-TBE gel using SYBRGold staining and a blue light transilluminator for visualization. After gel elution and reconstitution in water, 100 ng tRNA was transferred to a PCR tube and ligated to 20 pmol pre-adenylated adapter (A1, A2, A3, or A4) in 25% PEG-8000, 1xT4 RNA ligase buffer using 1 µL T4 RNA ligase 2 (truncated KQ), and 1 µL SuperaseIn. Prior to ligation adapters were adenylated using the NEB 5′ DNA Adenylation Kit following the manufacturer's instruction. After purification, adapter adenylation was verified using differential gel migration. Ligation reactions were incubated 6 hr at 25°C, pooled by adapter barcode and purified, followed by isolation of the ligation product from unligated tRNA using a 10% Urea-TBE gel.

After gel elution and reconstitution in water, the RT-PCR was performed as described by *Behrens et al., 2021*, using a similar RT oligo but with an extra nine random nucleotides at the 5′ end to act as a UMI. After the RT-PCR incubation, the remainder of the sample processing follows the charge tRNA-Seq sample processing described below, including cDNA circularization, Illumina P7/P5 sequence attachment, and sequencing.

## Charge tRNA-Seq method optimization

Optimization of the oxidation, cleavage, and dephosphorylation, collectively called the Whitfeld reaction (*Whitfeld and Markham, 1953*), was done using oligos *E. coli* tRNA-Lys-UUU-CCA and *E. coli* tRNA-Lys-UUU-CC (*Supplementary file 1*; anticodon omitted from name below). Both oligos were gel-purified on a 10% Urea-TBE gel to resolve full length from truncated oligos. First, the time required for oxidation was tested, following the same quenching and borax-buffered high pH-induced cleavage used by *Evans et al., 2017*. For this, samples of 35 ng *E. coli* tRNA-Lys-CCA were prepared in 10 µL 100 mM sodium acetate (pH = 4.5) and used as substrate for the Whitfeld reaction conversion to *E. coli* tRNA-Lys-CC. Reaction progress was monitored on a 10% Urea-TBE gel by resolving the one nucleotide difference using the substrate, the product, and a 50/50 mix as markers. Also using this approach, we tested using lysine-induced cleavage (*Khym and Cohn, 1961*) by swapping the sodium borate used for cleavage with 1 M lysine (pH = 8). The cleavage step also includes deacylation and to verify the completeness of this, four samples of 10 µg whole-cell RNA were prepared in 10 µL 100 mM sodium acetate (pH = 4.5) and incubated with 50 µL 1 M lysine (pH = 8) at 45°C for 5, 30, 90, and 270 min. Then, 1 mL ice-cold 80% IPA containing 100 mM sodium acetate (pH = 4.5) was added, RNA was precipitated, washed twice, dried, and reconstituted in 10 µL 100 mM sodium acetate (pH = 4.5). These deacylated samples were then submitted to the charge tRNA-Seq sample processing described below, except using lysine at pH = 9.5 and 90 min incubation at 45°C to ensure complete deacylation. From this, incubation time in lysine (pH = 8) was chosen to be 4 hr. To compare the RNA integrity after cleavage with lysine vs. borax, samples of 10 µg whole-cell RNA were prepared in 10 µL 100 mM sodium acetate (pH = 4.5) and added 50 µL of either 1 M lysine (pH = 8) or 100 mM sodium borate (pH = 9.5). Tubes were incubated 45°C and samples taken at time 0, 1.5, 4, and 8 hr. RNA integrity was determined using TapeStation (high sensitivity RNA) and 10% Urea-TBE gel. Upon combining the steps of the Whitfeld reaction to a one-pot reaction a color change was observed after addition of lysine. To test the effect of the periodate quencher, 10 µL of freshly prepared 200 mM NaIO$_4$ in 100 mM sodium acetate (pH = 4.5) was quenched by 10 µL 1 M aqueous solution of either ethylene glycol (*Neu and Heppel, 1964*), glycerol (*Alefelder et al., 1998*), glucose (*Evans et al., 2017*), ribose (*Watkins et al., 2022*), or water (control) for 10 min at room temperature. Then 100 µL 1 M lysine at either pH 8 or 9.5 was added and reactions incubated at 45°C for 4 hr before moving to room temperature for visual inspection (*Figure 2—figure supplement 1*, panel C).

For ligation optimization human tRNA was isolated from H1299 cells. First, whole-cell RNA was isolated as described above, reconstituted in water and deacylation at 45°C in 1 M lysine (pH = 8) for 4 hr. Then RNA was purified using the Monarch RNA Cleanup Kit (50 µg) and run on a 10% Urea-TBE gel to resolve the tRNA from mRNA and rRNA. tRNA was defined as the range between 70 and 85 nt as approximated by the low-range ssRNA ladder. For blunt-end ligations in *Figure 2—figure supplement 3*, 40 ng tRNA, either isolated from H1299 cells or as *E. coli* tRNA-Lys-CC oligo, was ligated to 20 pmol pre-adenylated adapter in a 20 µL reaction containing 25% PEG-8000, 200 U T4 RNA ligase 2 (truncated KQ; Rnl2tr KQ), 10 U SUPERaseIn, and the vendor provided buffer. For splint-assisted ligation in *Figure 2—figure supplement 4*, 35 ng tRNA, either isolated from H1299 cells or as *E. coli* tRNA-Lys-CC oligo, was ligated to 20 pmol annealed adapter:splint partial duplex as described for charge tRNA-Seq sample processing below. For the non-complementary splint test, two splint oligos were made with CAAC and AAC overhangs (*Supplementary file 1*) and annealed to adapter l1Sp. For the ligation test in *Figure 2*, panel E and *Figure 2—figure supplement 5*, panel A, 500 ng tRNA isolated from H1299 cells was subjected to the one-pot Whitfeld reaction described for charge tRNA-Seq sample processing below but with a single step removed. For the no oxidation sample NaIO₄ was replaced with NaCl, for the no dephosphorylation sample shrimp alkaline phosphatase (rSAP) was replaced with water, and for the no cleavage sample RNA was purified after periodate quenching. These were compared to a sample processed as described in *Evans et al., 2017*. All samples were purified using the Monarch RNA Cleanup Kit and 35 ng was used per ligation test with adapters l1Sp, l2Sp, and l3Sp using the ligation protocol described for charge tRNA-Seq sample processing below.

## Charge tRNA-Seq sample processing

Stepwise description with details given in *Supplementary file 2*. Whole-cell RNA was reconstituted in 100 mM sodium acetate (pH = 4.5) and keep on ice until the end of the periodate oxidation step. For deacylated control samples, RNA was prepared by first performing a deacylation step on the input RNA by incubation in 1 M lysine (pH = 8) at 45°C for 4 hr, followed by purification using the Monarch RNA Cleanup Kit (50 µg). The RNA concentration was adjusted to 1 µg/µL, 10 µL was transferred to a fresh tube, and 1 µL *E. coli* tRNA spike-in control was added. Initially, the spike-in control contained 5 ng/µL *E. coli* tRNA-Lys-CCA, later 5 ng/µL of each *E. coli* tRNA-Thr-CGT CCA-Phos and *E. coli* tRNA-Thr-CGT CCA was also included. To this 5 µL freshly prepared 200 mM NaIO₄ was added following 10 min incubation on ice, in the dark. For non-oxidized control samples, NaCl was used instead of NaIO₄. The oxidation was quenched by adding 5 µL 50% (vol/vol) ethylene glycol (~9 M) and incubating for 5 min on ice and 5 min at room temperature, in the dark. Then 50 µL 1 M lysine (pH = 8) with 1 µL SuperaseIn was added and tubes were incubated for 4 hr at 45°C. To dephosphorylate RNA 8 µL 10× rCutSmart Buffer and 1 µL rSAP was added followed by 30 min incubation at 37°C. RNA was then purified using the Monarch RNA Cleanup Kit (50 µg), eluting with 30 µL water. A 6 µL sample was then denatured by mixing with 2× urea loading buffer (8 M urea, 30 mM sodium acetate, 2 mM EDTA, 0.02% [wt/vol] bromophenol blue and xylene cyanol, pH adjusted to 4.7–5) and incubating 2 min at 90°C. The tRNA fraction was then isolated on a 10% Urea-TBE gel using SYBRGold staining and a blue light transilluminator for visualization. Gel elution was done by crushing the gel with a disposable pestle, adding 200 µL gel elution buffer and 1 µL SuperaseIn, then snap freezing in liquid nitrogen, and incubating at 65°C for 5 min with shaking. This gel slurry was filtered through a Spin-X filter followed by tRNA purification using the Oligo Clean & Concentrator kit. The concentration of purified tRNA was measured, then it was annealed in NEBuffer 2 by heating to 94°C for 2 min followed by cooling 1 °C/s to 4°C. 35 ng of the annealed tRNA was transferred to a PCR tube and to this was added 20 pmol annealed adapter:splint partial duplex, 1 µL 10× NEBuffer 2, 2 µL 10× T4 RNA ligase buffer, 4 µL 50% PEG-8000, 1 µL SuperaseIn, and 1 µL T4 RNA ligase 2. The annealed adapter:splint partial duplex was made by making an equimolar mix of the CCA and CC splint oligos, then using this to make an equimolar mix with the adapter oligo and annealing this in NEBuffer 2 by heating to 94°C for 2 min followed by cooling 0.3 °C/s to 4°C. Each ligation reaction was adjusted to 20 µL with water, mixed and incubated 1 hr at 37°C followed by 24 hr at 4°C and heat inactivation at 80°C for 5 min. Samples were pooled by adapter barcode, purified using the Oligo Clean & Concentrator kit, and then ligated tRNA was isolated on a gel and purified similarly to the initial tRNA isolation.

RT was set up with 60 ng of the purified adapter ligated tRNA as template using the buffer composition, incubation temperature, and time suggested by *Behrens et al., 2021*. To 10 µL template in a PCR tube, 2 µL 1.25 µM RT oligo and 4 µL RT buffer was added following denaturation and annealing by incubation at 90°C for 2 min, 70°C for 30 s and cooling 0.2 °C/s to 4°C. Then, to each tube 1 µL 100 mM DTT, 1 µL SuperaseIn, and 1 µL TGIRT-III RT polymerase (or Maxima H Minus for *Figure 2— figure supplement 6* and *Figure 2—figure supplement 7*) was added following 10 min incubation at 42°C. Then 1 µL 25 mM dNTPs was added and the incubation was resumed at 42°C for 16 hr on a thermocycler with the heated lid set to 50°C. The RNA template was hydrolyzed by adding 1 µL 5 M NaOH followed by incubation at 95°C for 3 min. The samples were then purified using the Oligo Clean & Concentrator kit and the cDNA was isolated on a gel and purified similarly to the initial tRNA isolation, eluting with 7 µL water. cDNA was circularized by transferring 5.5 µL cDNA to a PCR tube and adding 2 µL 5 M betaine, 1 µL 10× CircLigase buffer, 0.5 µL 1 mM ATP, 0.5 µL 50 mM MnCl$_2$, and 0.5 µL CircLigase. The reaction was incubated at 60°C for 3 hr on a thermocycler with a 70°C heated lid, then the enzyme was deactivated by denaturing at 80°C for 10 min.

PCR was used to attach Illumina P7/P5 sequences to flank the tRNA insert. Each PCR was set up to contain 0.6 µL circularized cDNA, 1.5 µL 10 mM dNTPs, 5 µL 10 µM P7 oligo, 5 µL 10 µM P5 oligo, 10 µL 5× KAPA HiFi buffer, 1 µL KAPA HiFi polymerase, and 26.9 µL water. The PCRs were incubated at 95°C for 3 min followed by 3 cycles of 98°C for 20 s, 68°C for 10 s, and 72°C for 15 s, and then followed by *X* cycles of 98°C for 20 s and 72°C for 15 s, with *X* being empirically determined (*Figure 2—figure supplement 10*, panel A). The optimal number of PCR cycles was determined by preparing three PCRs, incubating them with *X*=10, 12, and 14 and running 4 µL of each reaction on a 4–12% TBE gel. The PCRs with optimal *X*, resulting in abundant amplification product with little PCR crossover, were purified using the DNA Clean & Concentrator-5 kit and resolved on a 4–12% TBE gel. The gel was stained using SYBRGold and visualized using a blue light transilluminator to isolate the library DNA by cutting out the size range covering all possible insert lengths (170–290 bp). Gel elution was done by crushing the gel with a disposable pestle, adding 300 µL TBE, snap freezing in liquid nitrogen, and incubating at room temperature overnight with mixing. If necessary, elution time could be decreased by incubation at higher temperature, although this required adding higher salt concentrations to prevent DNA reannealing (*Figure 2—figure supplement 10*, panel B). The gel slurry was filtered through a Spin-X filter following DNA purification using the DNA Clean & Concentrator-5 kit and eluting with 20 µL 10 mM Tris (pH = 8). DNA with different Illumina P7/P5 barcodes were pooled for multiplexing and sequenced using Illumina paired-end sequencing using 2×100 bp reads.

## *E. coli* tRNA spike-in control

An *E. coli* tRNA spike-in control was generated from oligos *E. coli* tRNA-Lys-UUU-CCA and *E. coli* tRNA-Thr-CGT-CCAA (anticodon sometimes omitted from name). First, 2 µg per well of the *E. coli* tRNA-Lys-CCA oligo was loaded on a 10% Urea-TBE gel to resolve full length from truncated oligos. After gel elution and purification using the Oligo Clean & Concentrator kit the RNA concentration was measured and adjusted such that 5 ng was spiked into each sample of 10 µg whole-cell RNA before periodate oxidation. Adding the control before periodate oxidation afforded an internal control of the completeness of the oxidation reaction.

Second, 30 µL of 100 µM *E. coli* tRNA-Thr-CCAA oligo was submitted to a partial Whitfeld reaction, stopping before the dephosphorylation step. The oxidation reaction was performed by adding 10 µL 100 mM sodium acetate (pH = 4.5) and 20 µL 200 mM NaIO$_4$ followed by incubation for 30 min at room temperature in the dark. Oxidation was quenched using 20 µL 50% ethylene glycol and incubated 30 min at room temperature in the dark. Then buffer exchange was performed using a P-6 gel column pre-equilibrated with 100 mM lysine (pH = 8). To the eluate 400 µL 1 M lysine (pH = 8) and 1 µL SuperaseIn was added followed by 5 hr incubation at 45°C and purification using the Monarch RNA Cleanup Kit (using two 50 µg columns). The product, a 1 nt truncated and 3′ phosphorylated oligo named *E. coli* tRNA-Thr-CCA-Phos, was resolved on a gel to isolate the full-length oligo, as described for the other control. Half of this product was submitted to dephosphorylation using rSAP and purified using the Oligo Clean & Concentrator kit yielding *E. coli* tRNA-Thr-CCA. Complete phosphorylation of *E. coli* tRNA-Thr-CCA-Phos and complete dephosphorylation of *E. coli* tRNA-Thr-CCA were verified using ligation (*Figure 5—figure supplement 3*, panel A). Then concentrations of both *E. coli* tRNA-Thr-CCA-Phos and *E. coli* tRNA-Thr-CCA were measured to generate an equimolar mix adjusted such

that 10 ng was spiked into each sample of 10 µg whole-cell RNA before periodate oxidation. The 3′ phosphorylation protects from periodate oxidation and thus adding it before periodate oxidation afforded an internal control of a 50% charged tRNA, probing the completeness of the whole Whitfeld reaction and potential adapter ligation bias.

## Oligo design

For adapters used for blunt-end ligation the design was similar to *McGlincy and Ingolia, 2017*, with a 5′ phosphorylation to enable adenylation and a 3′ dideoxycytidine to prevent self-ligation and concatemer formation. For adapters A1, A2, A3, and A4 the barcode sequence was 8 nt starting at the 5′, for adapters l1N, l2N, and l3N the barcode sequence was truncated to 5 nt to make space for a preceding six random nucleotides to diversify the sequence context engaged in ligation.

The design of adapters used for splint-assisted ligation was influenced by *Smith et al., 2015*, and *Shigematsu et al., 2017*, but with several important differences listed below. First, we do not use ribonucleotides at any positions in our adapters or splint oligos. This affords us higher quality oligos due to the higher coupling efficiency of deoxyribose during oligo synthesis as well as robustness against hydrolysis of DNA compared to RNA. A primary reason to use ribonucleotides in the adapters and splint oligos is to increase ligation efficiency; however, we achieved ~100% ligation efficiency on isolated human tRNA using our design without ribonucleotides (*Figure 2—figure supplement 4*, panel A). Second, instead of ligating the adapter to the 3′ and the splint to the 5′ of the tRNA, we only ligate the adapter and block the splint from ligating using a 3′ C3 spacer, as well as dephosphorylating the 5′ of the tRNA. Similar to the blunt-end ligation adapters, a 3′ dideoxycytidine is included on all adapters to block self-ligation and concatemer formation. Third, we use two different lengths of splint oligos with overhang compatible with NCCA and NCC-ending tRNA. Fourth, our adapters vary in length by the size of their barcodes, from 5 to 8 nt. This is to offset the sequencing reading frame of read P2 (P7) as it progresses into the 3′ end of the tRNA, thus increasing the sequence diversity and base calling quality.

The RT-PCR oligo was designed in a similar way as *McGlincy and Ingolia, 2017*, with a 5′ phosphorylation for subsequent circular ligation of the cDNA and an 18-atom hexa-ethyleneglycol spacer (iSp18) to terminate the polymerase extension and avoiding rolling-circle amplification during the PCR to attach Illumina P7/P5 sequences. The RT oligo has a random purine base on the 5′ to increase circular ligation efficiency. We added an additional nine random nucleotides following this purine to increase the diversity of the sequence engaged in circular ligation. These random nucleotides also provide a UMI with 524,288 possible sequences that enable collapsing of reads derived from the same tRNA molecule. The UMI is also used as a general sample quality control by comparing the number of observed UMI sequences with the number expected. The expected number of unique UMI observations is calculated as:

$$E\left[X\right] = n\left[1 - \left(\frac{n-1}{n}\right)^{k}\right] \tag{1}$$

with $E[X]$ being the expected number of unique UMI observations, $n$ being the number of reads for the particular sample, and $k$ being the number of possible UMIs.

The final dsDNA library was designed as an Illumina TruSeq dual index library with combined i5 and i7 indices attached by PCR with P7/P5 oligos. These oligos were synthesized with a phosphorothioate bond between the last two nucleotides to prevent degradation by the KAPA HiFi polymerase. An overview of the RNA/DNA manipulations including ligation of adapters, RT-PCR, circularization, and library PCR is provided in *Supplementary file 3*.

## Read processing

Reads were first demultiplexed according to their i7/i5 barcodes. Read pairs were then trimmed and merged using AdapterRemoval:

```
AdapterRemoval --preserve5p --collapse --minalignmentlength 10 --adapter1
AGATCGGAAGAGCACACGTCTGAACTCCAGTCAC<P7_index >ATCTCGTATGCCGTCTTCTGCTTG
--adapter2 AGATCGGAAGAGCGTCGTGTAGGGAAAGAGTGT<P5_index >GTGTAGATCTCGGTGGTCGC
CGTATCATT --minlength<MIN_LEN>
```

with <P7_index> and <P5_index> defined by the i7/i5 index sequences for the given sample and <MIN_LEN> set to 25 for charge tRNA-Seq using blunt-end ligation and 39 for charge tRNA-Seq using splint-assisted ligation. Each file with merged reads were then split based on adapter barcode. A read was assigned to a particular adapter barcode if its 3' end had a substring within a hamming distance of one to the barcode sequence, including the region complementary to the splint. The adapter sequence was then trimmed off the 3' end; similarly, the 10 nt. UMI was located, saved, and trimmed off the 5' end, leaving only the tRNA sequence with possible 5' non-template bases introduced during RT-PCR. Finally, samples with an excess of 2e6 reads were downsampled to 2e6 reads.

Trimmed reads were aligned to a masked reference as described below using the Smith-Waterman algorithm implemented by SWIPE (**Rognes, 2011**):

```
swipe --symtype 1 --outfmt 7 --num_alignments 3 --num_descriptions 3
--evalue 0.000000001 --strand 1 -G 6 -E 3 --matrix<SCORE_MATRIX>'
```

With an input score matrix (<SCORE_MATRIX>) defining a match score of 1, a mismatch score of –3, and a score for alignment to a masked reference position (N) of 0.

Alignment results were processed to extract three key data: (1) tRNA charge, (2) relative expression level, and (3) mismatches, gaps, and RT truncations. First the alignment was parsed to extract transcript annotation(s), alignment score, and other relevant information. A read was assigned the annotation with the highest alignment score and upon ties up to three annotations were merged. When reporting data on the transcript level, a unique annotation was required for filtering, when reporting at the codon level multiple annotations were allowed but a unique anticodon was required and similarly for data on the amino acid level. Relative expression levels were calculated as RPM with a count correction such that reads with identical sequence and UMI were only counted once. Charge was calculated using uncorrected counts as this is a relative number. Mismatches, gaps, and RT truncations were extracted by redoing the Smith-Waterman alignment between the read and its unmasked transcript annotation using a match score of 1, a mismatch score of –2, a gap opening score of –3, and a gap extension score of –2. Using this new alignment, mismatched, gaps, and the index at the end of the alignment were extracted. Then for each transcript the fraction of reads having mismatches and gaps at a given position was calculated, additionally the percentage drop in coverage at each position is reported, here referred to as RT stops. For both mutation, gap fractions and RT stops, the UMI corrected read count was used. We provide a boilerplate example of the whole read processing workflow on GitHub: https://github.com/krdav/tRNA-charge-seq/blob/main/projects/example/process_data.ipynb.

## Reference masking

A human tRNA transcript reference for alignment was made from hg38 annotations in GtRNAdb (**Chan and Lowe, 2016**). These sequences were deduplicated and mitochondrial tRNAs and spike-in control sequences were appended. Then a BLAST database was generated, as required by SWIPE, using the makeblastdb application. To further improve the alignment specificity, a masked reference was made by converting positions with high likelihood of mismatch to Ns such that these have no negative contribution on the alignment score. Position-wise mismatch frequency was found as described above and filtered using a minimum of 200 transcript observations and 100 observations on each position. These were then turned into a masked reference using four tuning parameters for picking the positions to mask. unique_anno: Only count reads with a unique transcript annotation. min_mut_freq: The minimum mismatch frequency to trigger masking. frac_max_score: The minimum fraction of the maximum alignment score between two reference sequences to expand the masked positions in one reference to another, requiring both positions to have the same nucleotide and the acceptor position to have less than 100 observations. The purpose is for an abundant transcript to donate its masking to a highly similar, but less abundant, transcript likely having the same RNA modifications. iteration: The number of masking iterations to perform. When changing the reference for alignment by masking the annotations can change, thus changing the position-wise mismatch frequency and the resulting reference masking. Running multiple iterations of reference masking stabilizes this change.

To find the optimal combination of tuning parameters a grid search was performed, testing all combinations of parameters shown in **Figure 3**, panel A. The objective of the search is to minimize

the percentage of reads assigned to transcripts with multiple anticodons. Alternatively, the objective could be to minimize the percentage of reads assigned to multiple transcripts; however, this objective can lead the tuning parameters toward masking only a single transcript out of a family of highly similar transcripts, resulting in assignment of unique annotations to truncated reads, which cannot truly distinguish between transcripts of high similarity. This problem is less concerning using minimization of multiple anticodons since most families of highly similar transcripts have identical anticodons.

## Barcode replicate test

For the barcode replicate test shown in *Figure 4*, the RNA used was first incubated 8 hr at 20°C in intracellular physiological buffer, similar to the 8 hr timepoint described in the 'Aminoacylation half-life' section below. This provided tRNA containing a spectrum of charge levels, spanning from almost fully acylated isoleucine tRNAs to almost fully deacylated asparagine tRNA. A single 10 µg sample of this RNA was then subjected to the one-pot Whitfeld reaction and subsequent tRNA isolation and ligation to each of the nine adapters as described for charge tRNA-Seq sample processing above.

## Charge titration test

Whole-cell RNA was reconstituted with 100 mM sodium acetate (pH = 4.5) and adjusted to 1 µg/µL while keeping the RNA cold throughout. Half of this was moved to a fresh tube and deacylated by adding 5× volumes of 1 M lysine (pH = 8), incubating at 45°C for 4 hr, and purifying using the Monarch RNA Cleanup Kit. Meanwhile, the other half was stored at –80°C. The concentration of the deacylated RNA was adjusted to 1 µg/µL and mixtures of intact and deacylated RNA were made using the following percentages of intact/deacylated RNA: 100/0, 85/15, 70/30, 55/45, 40/60, 25/75, 10/90, 0/100. Then these mixtures were subjected to the charge tRNA-Seq sample processing protocol described above with between four and eight barcode replicates across independently prepared sequencing libraries, sequenced on different flow cells.

Reads were processed and the aminoacylation charge of each transcript was extracted to relate the measured with the predicted charge. However, the actual mixing ratios may deviate from the ones noted above due to inaccuracies in measuring the RNA concentration of intact and deacylated RNA, and due to depletion of certain tRNA species during the deacylation process, e.g., tRNAs sensitive to hydrolysis or depurination. We address this using a correction factor, $F_i$, described below. To calculate the predicted charge let $A$ represents intact RNA, $B$ represents deacylated RNA, and the index $i$ represents the transcript. Now, define the concentration, $C$, of a tRNA transcript $i$ in the intact RNA as 1, while letting the concentration of the same tRNA transcript in the deacylated RNA be a fraction, $F_i$, of the intact RNA:

$$C_i^A = 1$$

$$F_i = \frac{C_i^B}{C_i^A} <=> C_i^B = F_i \tag{2}$$

Then, define $T_i^A$ as the measured charge of the intact tRNA of a transcript $i$ averaged over the replicates, and similarly $T_i^B$ for deacylated RNA:

$$T_i^A = \text{Avg charge}\left(A_i\right)$$

$$T_i^B = \text{Avg charge}\left(B_i\right) \tag{3}$$

Now, the predicted charge of a mixture of $A$ and $B$ can be defined using $p$ to describe the percentage of $A$ in the mixture:

$$T_i^{AB}\left(p\right) = \frac{pT_i^A + \left(100 - p\right)T_i^B F_i}{p + \left(100 - p\right)F_i} \tag{4}$$

In the above, only $F_i$ is unknown. The titration was made with eight different mixing ratios, two of which are used to calculating $T_i^A$ and $T_i^B$, thus leaving six mixing ratios, each with several barcode replicates, to fit $F_i$. Fitting was performed by minimizing the sum of squared differences between predicted and measured charge using the Broyden-Fletcher-Goldfarb-Shanno (BFGS) algorithm with upper and lower bound constraints of 4 and 0.25. Then *Equation 4* was used to calculate the predicted charge

and the difference to the measured charge was found and broken down by adapter barcode to investigate ligation bias.

## Aminoacylation half-life

Whole-cell RNA was reconstituted with 1 mM sodium acetate (pH = 4.5) and adjusted to 1.5 μg/μL while keeping the RNA cold throughout. A zero timepoint was then taken and 80 μL was transferred to a PCR tube after which the experiment was started by adding 20 μL room temperature 5× buffer, quickly mixing, and placing the tube on a thermocycler set to 20°C. The buffer used was an intracellular physiological buffer at 1× containing: 19 mM NaCl, 125 mM KCl, 0.33 mM $CaCl_2$, 1.4 mM $MgCl_2$, 0.5 mM spermidine, 30 mM HEPES, adjusted to pH = 7.2 with KOH. Time from start of incubation was tracked and samples drawn at the following timepoints: 4 min, 8 min, 16 min, 32 min, 1 hr, 2 hr, 4 hr, 8 hr, 16 hr, and 40 hr. For the 40 hr timepoint, two samples were drawn: one standard and one receiving sham (NaCl) oxidation. Sample were taken by removing 8 μL, mixing it in a prepared tube with 2 μL ice-cold 500 mM sodium acetate (pH = 4.5) and storing it at –80°C until all timepoints were collected. This was repeated four times to generate independent replicates. Then samples were processed similar to the charge tRNA-Seq protocol described above, but with the three 5 min incubation times during periodate oxidation and quenching increased to 30 min each due to the lower periodate solubility in the presence of potassium ions.

After read processing and alignment, data integrity was verified by checking that the *E. coli* tRNA spike-in control and the non-oxidized 40 hr samples conformed to expectations. RNA integrity at the end of the experiment was also verified on a gel (*Figure 6—figure supplement 1*, panel A). The aminoacylation charge was then calculated at the codon level and the data fitted to an equation describing first-order decay:

$$N(t) = N_0 \left(\frac{1}{2}\right)^{\frac{t}{t_{1/2}}} + N_\infty \tag{5}$$

where $N(t)$ is the charge of a given codon as a function of time, $N_0$ is the charge at time zero, and $t_{1/2}$ is the decay half-life. We added the $N_\infty$ parameter to model the lower asymptote of charge to accommodate the small fraction of tRNAs that still presents with a CCA-end after full deacylation. The three parameters were fitted to the data by minimizing the sum of squared errors using the BFGS algorithm with upper and lower bound constraints for $N_0$ between 100% and 0%, for $t_{1/2}$ between 1e5 and 1 min, and for $N_\infty$ between 3.5% and 0%. A point estimate for the three parameters was found using all timepoints and replicates and a 95% confidence interval was found using bootstrapping ($N$=1000) by sampling a single time-series made up of random draws from the replicates (*Supplementary file 5*).

## Acknowledgements

We would like to acknowledge Arvind Rasi Subramaniam for suggesting us to set up charge tRNA-Seq, Alicia Darnell for sharing relevant samples and data, and David Sokolov for help with early method optimization. LBS acknowledges support from the National Institute of General Medical Sciences (NIGMS; R35GM147118).

## Additional information

### Funding

| Funder | Grant reference number | Author |
| --- | --- | --- |
| National Institute of General Medical Sciences | R35GM147118 | Lucas B Sullivan |

The funders had no role in study design, data collection and interpretation, or the decision to submit the work for publication.

## Author contributions
Kristian Davidsen, Conceptualization, Resources, Data curation, Software, Formal analysis, Validation, Investigation, Visualization, Methodology, Writing - original draft, Writing - review and editing; Lucas B Sullivan, Conceptualization, Supervision, Funding acquisition, Writing - original draft, Project administration, Writing - review and editing

## Author ORCIDs
Kristian Davidsen ⓘ https://orcid.org/0000-0002-3821-6902
Lucas B Sullivan ⓘ https://orcid.org/0000-0002-6745-8222

Reviewer #1 (Public Review): https://doi.org/10.7554/eLife.91554.3.sa1
Reviewer #2 (Public Review): https://doi.org/10.7554/eLife.91554.3.sa2
Author response https://doi.org/10.7554/eLife.91554.3.sa3

---

# Additional files

## Supplementary files
- Supplementary file 1. Oligo and reagents list.
- Supplementary file 2. Step-by-step transfer RNA sequencing (tRNA-Seq) protocol.
- Supplementary file 3. Overview of the RNA/DNA manipulation steps.
- Supplementary file 4. TGIRT versus Maxima read mapping statistics.
- Supplementary file 5. Transfer RNA (tRNA) aminoacylation half-life fitting data.
- MDAR checklist

## Data availability
Raw data and code for processing and recreating plots were uploaded to Zenodo: https://doi.org/10.5281/zenodo.8200906. Python code is available on Github: https://github.com/krdav/tRNA-charge-seq (copy archived at *Davidsen, 2024*). All other data is provided in the manuscript and supporting files.

The following dataset was generated:

| Author(s) | Year | Dataset title | Dataset URL | Database and Identifier |
| --- | --- | --- | --- | --- |
| Davidson K, Sullivan LB | 2024 | A robust method for measuring aminoacylation through tRNA-Seq | https://doi.org/10.5281/zenodo.10778311 | Zenodo, 10.5281/zenodo.10778311 |

The following previously published dataset was used:

| Author(s) | Year | Dataset title | Dataset URL | Database and Identifier |
| --- | --- | --- | --- | --- |
| Evans M, Clark W, Zheng G, Pan T | 2017 | Determination of tRNA aminoacylation levels by high throughput sequencing | https://www.ncbi.nlm.nih.gov/geo/query/acc.cgi?acc=GSE97259 | NCBI Gene Expression Omnibus, GSE97259 |

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
