## [Editor Report · eLife assessment]

This **valuable** paper presents a new protocol for quantifying tRNA aminoacylation levels by deep sequencing. The improved methods for discrimination of aminoacyl-tRNAs from non-acylated tRNAs, more efficient splint-assisted ligation to modify the tRNAs' ends for the following RT-PCR reaction, along with the use of an error-tolerating mapping algorithm to map the tRNA sequencing reads provide new tools for anyone interested in tRNA concentrations and functional states in different cells and organisms. The results and conclusions are **solid**, with well-designed tests to optimize the protocol under different conditions.

---

## [Referee Report · Reviewer #1 (Public Review)]

The manuscript of Davidsen and Sullivan describes an improved tRNA-seq protocol to determine aminoacyl-tRNA levels. The improvements include: (i) optimizing the Whitfeld or oxidation reaction to select aminoacyl-tRNAs from oxidation-sensitive non-acylated tRNAs; (ii) using a splint-assisted ligation to modify the tRNAs' ends for the following RT-PCR reaction; (iii) using an error-tolerating mapping algorithm to map the tRNA sequencing reads that contain mismatches at modified nucleotides.

The revised manuscript of Davidsen and Sullivan has addressed my concerns in the previous review. The authors performed a end-to-end comparison, which I requested - Fig. 2 and Fig S2. This is exactly what I meant, albeit the differences in each method to perform the comparison of the detectability. The manuscript is a strong methodological improvement of the tRNA quantification protocols!

---

## [Referee Report · Reviewer #2 (Public Review)]

Davidsen and Sullivan present an improved method for quantifying tRNA aminoacylation levels by deep sequencing. By combining recent advances in tRNA sequencing with lysine-based chemistry that is more gentle on RNA, splint oligo-based adapter ligation, and full alignment of tRNA reads, they generate an interesting new protocol. The lab protocol is complemented by a software tool that is openly available on Github. Many of the points highlighted in this protocol are not new, but have been used in recent protocols such as Behrens et al. (2021) or McGlincy and Ingolia (2017). Nevertheless, a strength of this study is that the authors carefully test different conditions to optimize their protocol using a set of well-designed controls.

The conclusions of the manuscript appear to be well supported by the data presented. However, the lack of benchmarking relative to other methods remains as a key criticism also after this revision.

(1) The manuscript reports a different method to measure aminoacylation of tRNA. The main point that remains unsatisfactory is a better benchmarking of such aminoacylation measurements against the state of the art. In the current form of the revised manuscript it is not possible to estimate how much the results of this new protocol differ from alternative methods and in particular from Behrens et al. (2021). Here it will be helpful to perform experiments with samples similar to those (like HEK cells or yeast cells) used in the mim-tRNAseq study and not with H1299 cells.

The claim that a comparison to every published protocol is not feasible is not a good argument for not performing any benchmarking experiments. Such benchmarking experiments are not meant to define the ground truth but are needed to estimate the difference in the outcome of different protocols. I agree with the authors that precision/reproducibility is essential when developing a new protocol. But the analysis and comparison should not stop there.

(2) The reported protocol can not only be used for quantification of tRNA aminoacylation but it can also be used for tRNA quantification and analysis of tRNA modifications. It will increase the impact of this study if the authors benchmark the outcomes of their protocol with other tRNA sequencing protocols with samples similar to these papers, which will be important for certain research teams that are unlikely to implement two different tRNA sequencing methods.

The authors decided not to perform further experiments in cell lines or mutants that allow a comparison to other published methods. In my opinion this limits the impact of the work. But as a reviewer I can only make recommendations. It is the authors decision to take those or not.

---

## [Author Response]

The following is the authors’ response to the original reviews.

**eLife assessment**
This valuable paper presents a new protocol for quantifying tRNA aminoacylation levels by deep sequencing. The improved methods for discrimination of aminoacyl-tRNAs from non-acylated tRNAs, more efficient splint-assisted ligation to modify the tRNAs' ends for the following RT-PCR reaction, and the use of an error-tolerating mapping algorithm to map the tRNA sequencing reads provide new tools for anyone interested in tRNA concentrations and functional states in different cells and organisms. The results and conclusions are solid with well-designed tests to optimize the protocol under different conditions.
**Public Reviews:**

We thank both reviewers for suggestions, feedback and improvements. We address these pointwise below.

**Reviewer #1 (Public Review):**
Summary:The manuscript of Davidsen and Sullivan describes an improved tRNA-seq protocol to determine aminoacyl-tRNA levels. The improvements include: (i) optimizing the Whitfeld or oxidation reaction to select aminoacyl-tRNAs from oxidation-sensitive non-acylated tRNAs; (ii) using a splint-assisted ligation to modify the tRNAs' ends for the following RT-PCR reaction; (iii) using an error-tolerating mapping algorithm to map the tRNA sequencing reads that contain mismatches at modified nucleotides.Strengths:The two steps, the oxidation, and the splint-assisted ligation are yield-diminishing steps, thus the protocol of Davidsen and Sullivan is an important improvement of the current protocols to enhance the quantification of aminocyl-tRNAs.Weaknesses:The oxidation and the selection of aminoacyl-tRNA is the first step in all protocols. Thereafter they differ on whether blunt ligation, hairpin (DM-tRNA-seq, YAMAT-seq, QuantM-seq, mim tRNA-seq, LOTTE tRNA-seq), or splint ligation is used and finally what detection method is applied (i-tRAP, tRNA microarrays). What is the correlation to those alternative approaches (e.g. i-tRAP (PMID 36283829), tRNA microarrays (PMID: 31263264) etc.)? What is the correlation with other approaches with which this improved protocol shares some steps (DM-tRNA-seq, mim-tRNA-seq)?

We appreciate the fair assessment and fully agree that our work would benefit from a large comparison between all known tRNA-seq methods. We did directly compare many elements of our method to those of other methods (e.g. ligation efficiency and barcode bias); however, as noted by the reviewer we did not perform a direct end-to-end comparison with all other methods. An ideal comparison would require running several different sample conditions and technical replicates through our protocol and repeating the process across a half dozen or so other methods as they are described. Unfortunately, this approach is unlikely to be feasible since each method uses different oligos, reagents and kits, and all would have to be acquired at substantial cost. Some methods also rely on other detection methods such as microarrays, qPCR, or Illumina sequencing, which would also make this goal all the more onerous. There are also different pipelines for data processing that, in some instances, make the final results hard to compare. In short, this would be a monumental and expensive task to do comprehensively. We also worry that, even if these experiments were conducted such that some variables were concluded to be superior, they could still be challengeable based on perceived or actual protocol differences from the prior art. In summary, we think that an overall comparison with each method would be ideal, but practical concerns limit us to optimizing and comparing the variables that we found to be most prone to introducing bias in the results.

For methods that measure tRNA expression levels (DM-tRNA-seq, YAMAT-seq, QuantM-seq, mim-tRNA-seq, LOTTE tRNA-seq etc.) there are some fundamental problems regarding absolute quantification using NGS that preclude simple comparisons. These problems are well known in the field of microRNA (Fuchs et al. (2012) [PMID: 25942392]) and arise due to several factors introduced during processing steps such as purification, ligation, reverse transcription and amplification. With the lack a “true” quantitation benchmark it would be difficult to make quantitative claims from each. Therefore, in our own work we benchmark tRNA expression levels for sample-to-sample reproducibility (i.e. precision) as further explained in the response to reviewer #2.

For comparison to methods that measure tRNA charge we did have an opportunity to compare our results with those of another study. To this end, we have added a figure comparing the baseline charge found using our method and the one used in Evans et al. (Revised manuscript Figure 2—figure supplement 9). This comparison finds broadly similar results for tRNA charge, including similar trends for a subset of Glu, Ser and Pro codons that are notable for their lowered basal tRNA charge.

**Reviewer #2 (Public Review):**
Davidsen and Sullivan present an improved method for quantifying tRNA aminoacylation levels by deep sequencing. By combining recent advances in tRNA sequencing with lysine-based chemistry that is more gentle on RNA, splint oligo-based adapter ligation, and full alignment of tRNA reads, they generate an interesting new protocol. The lab protocol is complemented by a software tool that is openly available on Github. Many of the points highlighted in this protocol are not new but have been used in recent protocols such as Behrens et al. (2021) or McGlincy and Ingolia (2017). Nevertheless, a strength of this study is that the authors carefully test different conditions to optimize their protocol using a set of well-designed controls.The conclusions of the manuscript appear to be well supported by the data presented. However, there are a few points that need to be clarified.

We appreciate the acknowledgement of the strength of our aminoacylation controls and agree that our method is relying on many aspects of the mentioned prior work.

(1) One point that remains unsatisfactory is a better benchmarking against the state of the art. It is currently impossible to estimate how much the results of this new protocol differ from alternative methods and in particular from Behrens et al. (2021). Here it will be helpful to perform experiments with samples similar to those used in the mim-tRNAseq study and not with H1299 cells.

We fully agree that more rigorous benchmarking would be desirable. As also noted in the response to reviewer #1, a full end-to-end comparison of methods would be ideal but would be onerous and expensive in practice, so we focused on optimizing the steps we found to be most prone to introducing bias in the data.

We agree that Behrens et al., (2021) has substantial methodological overlap with our work and was instrumental in our efforts; however, the focus of their manuscript was largely on quantification of tRNA abundance and modifications, rather than the tRNA charge. In fact, tRNA charge was only determined for yeast in that study. Quantifying the abundance of short RNAs using NGS is very difficult (Fuchs et al. (2012) [PMID: 25942392]) and will likely require the use of a mixture of tRNAs as spike-in references for normalization (Bissels et al. (2009) [PMID: 19861428]). In the case of Behrens et al. (2021), they did not use a spike-in tRNA reference, but instead correlated gene copy number with their measured tRNA abundance. They also compare to Northern blotting for two tRNA transcripts, showing a directionally similar result; however, no quantitative claims can be made measurement accuracy. Until a good method of normalizing tRNA quantification is found, we believe that sample-to-sample reproducibility (i.e. precision) is the most useful objective to optimize because this will allow detection of differential expression. Towards that end, we quantified the precision of our method (Figure 4 and its two supplementary figures) with associated statistics, which can be used to estimate the number of samples required to detect significance during differential expression analysis. For tRNA charge, quantification is easier, which is why we present statistics on both accuracy and precision. In this case we can better compare results across methods, and so we have added a comparison of our results to the charge quantification from Evans et al. (2017) (Figure 2—figure supplement 9).

(2) While the protocol aims to implement an improved method for quantification of tRNA aminoacylation, it can also be used for tRNA quantification and analysis of tRNA modifications. It will increase the impact of this study if the authors benchmark the outcomes of their protocol with other tRNA sequencing protocols with samples similar to these papers, which will be important for certain research teams that are unlikely to implement two different tRNA sequencing methods. Are there any possible adaptations that would allow the analysis of tRNA fragments?

The first part of this comment regarding comparison of methods is addressed in response to in the prior reviewer comment and in the response to reviewer 1. In the specific case of tRNA modifications, the issue is similar to abundance quantification in that a “true” reference of modified tRNA is likely necessary for proper quantification, alongside testing of each method simultaneously.

Regarding tRNA fragments, our method is not suitable for this use case. This is because our adapter ligation step depends on an intact tRNA structure with either CCA or CC overhang on the 3’-end and thus we almost exclusively get reads with CCA/CC ends and no reads from fragments. This specificity is good for increasing charge quantification accuracy but not good for the methods versatility. For a more versatile method we recommend Watkins et al. (2022) [PMID: 35513407].

(3) Like Behrens et al. (2021), Davidsen and Sullivan use TGIRT-III RT for their analyses. The enzyme is not currently available in a form suitable for tRNA-seq. It would be very helpful to test different new RT enzymes that are commercially available. The example of Maxima RT - Figure 2 Supp 6 - shows significantly lower performance than the presented TGIRT-III RT data. In lines 296-298, the authors mention improvements to the protocol by using ornithine. Why are these improvements not included?

We share similar concerns that the TGIRT-III enzyme is no longer commercially available. It became unavailable while we were preparing this manuscript, reflected by the fact that almost all our figures are made using this enzyme. Others have discovered this too and Lucas et al. (2023) [PMID: 37024678] tested several RT polymerases using TapeStation as a readout for readthrough. As they reported that Maxima has good performance, we decided to test it on a full run with replicates. The results are outlined in Figure 2—figure supplement 6 and for resubmission we have added a table to the appendix that compares the alignment statistics. Unfortunately, the readthrough of the Maxima polymerase on cytoplasmic tRNAs is not as high as for TGIRT-III; however, interestingly it seems to have better performance for mitochondrial tRNAs (Figure 2 – Figure Supplement 6). Regardless, in the initial paper submission we failed to evaluate whether this readthrough difference affected charge measurements. We have now fixed this by adding Figure 2—figure supplement 7, which shows that there are no differences in charge measurements TGIRT-III vs. Maxima. Not surprisingly, there are substantial differences between polymerases when looking at relative tRNA abundance (*which affirms the discussion above related to the difficulty of tRNA abundance quantification*); however, the high sample-to-sample reproducibility remains intact with either polymerase. An exhaustive search for better polymerases is warranted but falls outside the scope of our work.

Regarding the improvements suggested by us, using ornithine as a cleavage catalyst instead of lysine, we first learned about this possibility later and thus only want to make readers aware that other options exist. We have clarified the paragraph to make this clearer.

(4) A technical concern: The samples are purified multiple times using a specific RNA purification kit. Did the authors test different methods to purify the RNA and does this influence the result of the method?

In the past, we have relied exclusively on alcohol precipitation but during the development of this protocol we found it easier and more reproducible to use column-based purification when possible. However, as we have not made a direct comparison this remains anecdotal evidence. Nonetheless, to minimize any possible bias of column-based purification you will notice that we use columns with binding capacity 5x higher than the highest amount of RNA/DNA added to the column.

(5) The study would benefit from an explicit step-by-step protocol, including the choice of adapters that are shown to work best in the protocol.

This is a great point! We have included tables with all the oligos used (Supplementary file 1), a detailed step-by-step protocol with pictures of anticipated gel results (Supplementary file 2) and an overview of the RNA/DNA manipulations to make it clear where adapter sequences are located (Supplementary file 3). For the data processing we provide a comprehensive example in the Github repository. All this was included in our first submission of this manuscript (as well as on bioRxiv), but we suspect this was not readily accessible to the reviewers. We will make sure that these documents are going to be available through eLife and have emphasized their existence in the main text of the manuscript.

**Recommendations for the authors:**

**Reviewer #1 (Recommendations For The Authors):**
To stratify this improvement a comparison to the most common methods should be made. For example, how do the results with the improved protocol with i-tRAP (PMID 36283829), tRNA microarrays (PMID: 31263264), or with the approaches the improved protocol shares with some other tRNA-seq approaches (DM-tRNA-seq, mim-tRNA-seq)?

Once again, we thank the reviewer for the good recommendations. The points about direct comparisons were discussed above.

**Reviewer #2 (Recommendations For The Authors):**

These are all great points; we address them below.

Minor points:- Please use chemical conventions, e.g. for mcm5s2U and NaIO4 with superscript or subscript.

Fixed.

- Figure 2F: Glu GAA is only 82% charged; can this be due to mcm5s2U (Figure 3 supp 2) leading to a misalignment? What happens to Ser-NNN? Why is mitochondrial tRNA so much less charged?

Regarding the Glu-GAA charge at baseline, we do not think this is an artifact of the mcm5s2U modification as it would then also be expected for Gln-CAA and Lys-AAA. The same occurs in the charge data in Evans et al. (2017) and they use a very different alignment strategy. Lastly, the charge titration and half-life experiments show no evidence of inaccuracy/bias for Glu-GAA.

But the question remains – why is the charge of Glu-GAA so low? At this point our best guess is speculative. It may have something to do with the strong enrichment of Glu-GAA codons in the A site found by ribosome profiling on mouse embryonic stem cells (Ingolia et al. (2011) [PMID: 22056041]).

- Spell out "clvg" or "dphs" in the figure legend of Figure 2 and others. Similar for other abbreviations in figures. They are not always explained in the legends.

Fixed.

- Figure 3 supp 2: Please use U instead of T in the anticodons. The labels are a bit confusing. Please clearly align to the tick (also for Figure 3C).

Fixed.

- Line 220-223. Which RT enzyme was used for Figure 3 supp 2? Does it make a difference?

TGIRT-III was used. Only Figure 2—figure supplement 6 and Figure 2—figure supplement 7 (added for resubmission) show data with the Maxima polymerase. To address the second part of the question we have added a comparison between TGIRT-III and Maxima for mcm5s2U modification detection (Figure 3—figure supplement 3). Interestingly, there is a polymerase specific signature for mcm5s2U modifications; however, more work would be required to determine which polymerase is best suited for detection of this and other modifications.

- Figure 4 supp 1 and Figure 4 supp 2 change order.

Fixed.

Typos:- Figure 1 and Figure 1-figure supplement 1: In the periodate the "-" is in a small box (at least in my PDF viewer). Can this box be removed?- Line 175: duplicated verb.- Line 348: "moved".

Thanks for catching these. They have now been fixed.